# On the stability of gradient descent with second order dynamics for time-varying cost functions

**Travis E. Gibson**[1,2,3,4,*], **Sawal Acharya**[2,5], **Anjali Parashar**[1], **Joseph E. Gaudio**[1], **and Anuradha M. Annaswamy**[1]

[1]Massachusetts Institute of Technology
[2]Brigham and Women's Hospital
[3]Harvard Medical School
[4]Broad Institute of MIT and Harvard
[5]Stanford University
[*]Correpsondence: tegibson@bwh.harvard.edu

**Reviewed on OpenReview:** https://openreview.net/forum?id=HlzjI2fn2T

## Abstract

Gradient based optimization algorithms deployed in Machine Learning (ML) applications are often analyzed and compared by their convergence rates or regret bounds. While these rates and bounds convey valuable information they don't always directly translate to stability guarantees. Stability and similar concepts, like robustness, will become ever more important as we move towards deploying models in real-time and safety critical systems. In this work we build upon the results in Gaudio et al. 2021 and Moreu & Annaswamy 2022 for gradient descent with second order dynamics when applied to explicitly time varying cost functions and provide more general stability guarantees. These more general results can aid in the design and certification of these optimization schemes so as to help ensure safe and reliable deployment for real-time learning applications. We also hope that the techniques provided here will stimulate and cross-fertilize the analysis that occurs on the same algorithms from the online learning and stochastic optimization communities.

## 1 Introduction

For data-driven technologies to be reliably deployed in the real world they will inevitably need robustness guarantees. Those guarantees, while practical, will also aid in the formal certification of these methods. Robustness guarantees will be even more important for Safety Critical and Real-Time Systems (SCS & RTS). Part of these robustness guarantees will assuredly be related to cross-validation. There are also many scenarios where we envision the optimization scheme itself would also need to have robustness guarantees. Here, specifically, we are interested in the robustness guarantees of Gradient Descent (GD) based optimization schemes. Consider for instance systems that learn in real-time while they are deployed, like a self-driving car that catalogues new events and simultaneously labels and retrains the model (semi-supervised) on the fly, or an aircraft control algorithm that updates the feedback control gains in flight. In both these instances one does not have the luxury of tuning learning rates offline or performing episodic learning and control, but needs to be assured that the model training process itself is stable in real-time. By stability here we are referring to the definitions from dynamical systems and control theory (Massera, 1956), not the algorithmic stability definition from Bousquet & Elisseeff (2002).

Within control this field of study falls under the moniker of Adaptive Control (AC).[1] The closest area within ML where models are trained in real-time is fairly nascent but aptly named Real-Time Machine Learning (RTML) (DARPA; NSF).[2] With these applications in mind we are interested in analyzing the stability of gradient descent algorithms with second order dynamics when applied to **time varying** cost functions. *To simplify notation we will refer to GD with second order dynamics as simply second order GD (not to be confused with the order of the gradients that are taken).*

GD and its variants have rightfully attracted significant attention given their success as the workhorses for training deep neural networks (Goodfellow et al., 2016). The accelerated gradient methods first introduced by Polyak (1964) and Nesterov (1983) have had renewed interest (Su et al., 2014; Wibisono et al., 2016), as well as the expanded adoption and innovation within stochastic gradient descent methods and online learning (Cesa-Bianchi & Lugosi, 2006; Shalev-Shwartz, 2012; Hardt et al., 2016; Bottou et al., 2018). The most popular optimization schemes for deep models combine ideas from stochastic gradient descent, the classic accelerated methods with momentum (Tieleman & Hinton, 2012), and adaptive gradient methods (Duchi et al., 2011), e.g. Adam (Kingma & Ba, 2015). Methods like Adam are analyzed with Regret Analysis (RA) which frames the analysis in terms of minimizing some accumulated cost over time as apposed to the direct analysis of something like a Lyapunov function. We note that there are other examples where Stochastic GD with Momentum (SGDM) are analyzed with Lyapunov functions (Liu et al., 2020), bringing those works more in line with the style of analysis presented in this work. See §4 for a detailed discussion about RA and SGDM.

Orthogonal work to the above, that also investigated higher order gradient descent techniques, occurred within the control theory community under the name of a **High-order Tuner (HT)** (Morse, 1992). Prior to the introduction of the HT by Morse, parameter updates used in model reference adaptive control were typically defined using only first order derivatives (i.e. if the adaptive parameter is $\theta(t)$ then the update would be $d\theta/dt = \ldots$). Morse showed how to construct update laws for adaptive control parameters where the time derivative could be any arbitrary order $n$, (i.e. $d^n\theta/dt^n = \ldots$). Note that the update law is what one uses to "tune" the adaptive control parameters, hence the name "high-order tuner".

Gaudio et al. (2021) and Moreu & Annaswamy (2022) recently showed that a HT could be parameterized to look like the popular accelerated methods first introduced by Nesterov (1983) and that in turn, one could study their stability when applied to **explicitly time varying functions**. Our work picks up from here. The contributions of this work include streamlined stability proofs for time varying smooth convex functions where prior art's proofs were for linear regression models only (Gaudio et al., 2021) or for more general functions but with hyper-parameters that were fixed in time and with exceedingly strict constraints (Moreu & Annaswamy, 2022). Specifically, our analysis demonstrates that a simpler discrete time parameterization of the HT is possible (coinciding with Nesterov form II as it is sometimes referred to (Ahn & Sra, 2022)), a different Lyapunov candidate can be used, and that together these allow for a less conservative bound on the (now time-varying) hyper-parameters.

This paper is organized as follows. In Section 2 we formally state our problem along with the major assumptions stated explicitly. In Section 3 we present our main theoretical results. Section 4 provides a detailed discussion around the literature. This is followed by Section 5 where we provide additional simulations. In Section 6 we discuss acceleration within the context of time varying objective functions. We close with our conclusions in Section 7.

## 2 Problem Statement and Update Rule

We consider the following optimization problem, $\min_{\boldsymbol{x}\in\mathbb{R}} f_t(\boldsymbol{x})$ where for every time $t$, $f_t : \mathbb{R}^N \to \mathbb{R}$. The subscript $t$ denotes the time variation in $f_t(\cdot)$ which could occur for a fixed cost function but with streaming

---

[1] Here we are referring to AC that is simultaneously updating parameters while controlling, like Model Reference Adaptive Control (MRAC) (Narendra & Annaswamy, 2012; Ioannou & Sun, 2012), not the area of AC that has the learning phase and the deployment phase separate, i.e. episodic learning (Hazan & Singh, 2022; Tsiamis et al., 2022).

[2] Continuous Learning (CL) and Lifelong Learning (LL) (Chen & Liu, 2018; Parisi et al., 2019) are similar in name to RTML, but CL and LL are primarily concerned with concepts like catastrophic forgetting (McCloskey & Cohen, 1989) with an eye toward learning new things without forgetting what you learned in the past.

data or in scenarios where the objective function itself changes over time. The algorithm we study in this report falls under the umbrella of what we are calling a High-order Tuner (HT) which are gradient descent algorithms with second order dynamics similar in structure to the popular momentum methods, ***but with explicitly time varying cost functions***.

The HT studied here is parameterized as follows

$$\boldsymbol{x}_t = \beta_t \boldsymbol{z}_t + (1 - \beta_t)\boldsymbol{y}_t \tag{1a}$$

$$\boldsymbol{y}_{t+1} = \boldsymbol{x}_t - \alpha_t \nabla f_t(\boldsymbol{x}_t) \tag{1b}$$

$$\boldsymbol{z}_{t+1} = \boldsymbol{z}_t - \eta_t \nabla f_t(\boldsymbol{x}_t) \tag{1c}$$

where $\alpha_t$, $\beta_t$, and $\eta_t$ are all (potentially) time varying hyper-parameters. This is almost identical to Nesterov (2018) with the significant difference being that the functions $f_t$ are explicitly time varying.

For ease of discussion we will use the following definitions

$$\alpha_t := \frac{\mu_t}{N_t} \qquad \eta_t := \frac{\gamma_t}{N_t} \qquad \bar{\beta}_t := 1 - \beta_t \qquad \bar{\mu}_t := 1 - \mu_t \tag{2}$$

where $\gamma_t$, $\mu_t$, $\beta_t$, and $N_t$ are the free design parameters. We will refer to $N_t$ as the normalization parameter. All norms, $\|\cdot\|$, are 2 norms. We will use the following definitions and assumptions throughout the paper.

**Definition 1** (convex). A function $f : \mathbb{R}^N \to \mathbb{R}$ is convex if $\forall \boldsymbol{x}, \boldsymbol{y} \in \mathbb{R}^N$ and $\lambda \in [0, 1]$,

$$f(\lambda \boldsymbol{x} + (1 - \lambda)\boldsymbol{y}) \leq \lambda f(\boldsymbol{x}) + (1 - \lambda)f(\boldsymbol{y}).$$

**Definition 2** (smooth). A continuously differentiable function $f : \mathbb{R}^N \to \mathbb{R}$ is L-smooth if there exists an $L < \infty$ such that $\forall \boldsymbol{x}, \boldsymbol{y} \in \mathbb{R}^N$, then following holds $\left\| \nabla f(\boldsymbol{x}) - \nabla f(\boldsymbol{y}) \right\| \leq L \|\boldsymbol{x} - \boldsymbol{y}\|$.

**Definition 3** (strong convexity). A differentiable function $f : \mathbb{R}^N \to \mathbb{R}$ is $\sigma$-strongly convex if $\forall \boldsymbol{x}, \boldsymbol{y} \in \mathbb{R}^N$ and $\sigma \geq 0$, then following holds $f(\boldsymbol{y}) \geq f(\boldsymbol{x}) + \nabla f(\boldsymbol{x})^\mathsf{T}(\boldsymbol{y} - \boldsymbol{x}) + \frac{\sigma}{2} \|\boldsymbol{y} - \boldsymbol{x}\|^2$.

**Assumption 1.** For each $t$ the function $f_t$ is an $L_t$-smooth convex function and furthermore the sequence $L_t$ is bounded, i.e. $L_t \in \ell_\infty$ (see Definitions 1 and 2).

**Assumption 2.** There exists an $\boldsymbol{x}^*$ such that $f_t(\boldsymbol{x}^*) = \min f_t(\boldsymbol{x})$ for all $t$.

**Assumption 3.** For each $t$ an upper bound on the smoothness of the function is obtained, $N_t \geq L_t$, for explicit use in the HT, see Equation (2). If $L_t$ is bounded then $N_t$ is bounded as well.

**Remark 1.** Assumptions 1 and 3 allow for arbitrary variation in the smoothness of the time varying function, but the smoothness, and therefore the bound on the smoothness, can not grow arbitrarily large. If we did allow arbitrarily large smoothness bounds then we will still be able to prove stability of our algorithm but not its asymptotic convergence. In order for Assumption 3 to be satisfied one will need a practical means of estimating the smoothness, $L_t$, of the function. For twice differentiable functions the trace of the Hessian is efficiently computable and can be used as a bound for the smoothness.[3] Finally, we note that in order to obtain asymptotic convergence guarantees we need the optimal point $\boldsymbol{x}^*$ to be fixed, but given that our results are grounded in a stability analysis framework (all signals remain bounded with arbitrary initial conditions), ***the optimal point can actually change over time and stability is maintained*** (see §3.4).

## 3 Stability for time varying functions

Our objective is to provide streamlined, tractable, and complete proofs that fit within the main text. We will not begin with the most general results first, instead starting with a less general result that is easier to follow, and then proceeding onto the general result. In section §3.1 we present our first result for smooth

---

[3]For twice differentiable functions and by the intermediate value theorem we have that $\nabla f(\boldsymbol{x}) - \nabla f(\boldsymbol{y}) = \nabla^2 f(\boldsymbol{z})(\boldsymbol{x} - \boldsymbol{y})$ for some $\boldsymbol{z} \in [\boldsymbol{x}, \boldsymbol{y}]$. The hessian $\nabla^2 f$ of a convex function is a positive semi-definite (symmetric) matrix, and so for any $z$ the induced 2-norm $\left\| \nabla^2 f(\boldsymbol{z}) \right\| = \lambda_{\max}(\nabla^2 f(\boldsymbol{z}))$ where $\lambda_{\max}$ denotes the largest eigenvalue. Computing the eigenvalues of a matrix requires full knowledge of the matrix, but if we simply want to bound the eigenvalues we can recall that the trace of a square matrix is equal to the sum of its eigenvalues. This taken together results in the bound $\left\| \nabla f(\boldsymbol{x}) - \nabla f(\boldsymbol{y}) \right\| \leq \sup_{\boldsymbol{c} \in [\boldsymbol{x}, \boldsymbol{y}]} \text{trace}(\nabla^2 f(\boldsymbol{c})) \|\boldsymbol{x} - \boldsymbol{y}\|$

convex functions (Proposition 1), with the more general results coming with Theorem 1 in §3.2 . Then in §3.3 we present our analysis for strongly convex functions with Proposition 2. For the strongly convex setting obtaining the exponential convergence rate in a simple form necessitates some degree of simplification so we use the simplified setting of Proposition 1 in our presentation of Proposition 2. A completely general bound for the strongly convex setting is straightforward but challenging to present in a compact form without the introduction of a significant number of intermediate variables. We close with a discussion on time varying optimal points.

### 3.1 Warm-up direct proof for smooth convex

**Proposition 1.** *Let $f_t$ satisfy Assumptions 1-3. For the HT defined in Equation (1) with arbitrary initial conditions $\{\boldsymbol{x}_0, \boldsymbol{y}_0, \boldsymbol{z}_0\}$, $N_t \geq L_t$ ($f_t$ is $L_t$ smooth by assumption), and the following three conditions satisfied*

$$\beta_t \in [0, 1] \tag{3a}$$

$$\mu_t \in [\epsilon, 1], \quad \epsilon > 0 \tag{3b}$$

$$\gamma_t = \tfrac{1}{2}\mu_t. \tag{3c}$$

*then*

$$V_t = \|\boldsymbol{z}_t - \boldsymbol{x}^*\|^2 + \|\boldsymbol{y}_t - \boldsymbol{z}_t\|^2 \tag{4}$$

*is a Lyapunov candidate and it follows that $V_t \in \ell_\infty$. Furthermore, $\lim_{t \to \infty} \left[ f_t(\boldsymbol{x}_t) - f_t(\boldsymbol{x}^*) \right] = 0$.*

*Proof.* To demonstrate that $V_t$ is bounded we will first show that $\Delta V_{t+1} = V_{t+1} - V_t \leq 0$. The basic strategy is to obtain a bound for $\Delta V_{t+1}$ in a quadratic form of the vectors $\nabla f_t(\boldsymbol{x}_t)$ and $(\boldsymbol{x}_t - \boldsymbol{z}_t)$, making as few intermediate bounds as possible until the quadratic form is obtained, for which we then complete the square so as to obtain constraints on the free design parameters. The final ingredient is a simple argument regarding bounded monotonic sequences.

We begin by expanding the time step ahead Lyapunov candidate $V_{t+1} = \|\boldsymbol{z}_{t+1} - \boldsymbol{x}^*\|^2 + \|\boldsymbol{y}_{t+1} - \boldsymbol{z}_{t+1}\|$ from (4) by substituting the update for $\boldsymbol{y}_{t+1}$ from (1b) and $\boldsymbol{z}_{t+1}$ from (1c) and noting that $\boldsymbol{x}_t - \boldsymbol{z}_t = \bar{\beta}_t(\boldsymbol{y}_t - \boldsymbol{z}_t)$ from (1a) where $\bar{\beta}_t = 1 - \beta_t$, as defined in (2), we obtain

$$V_{t+1} = \|\boldsymbol{z}_t - \boldsymbol{x}^*\|^2 + (\eta_t^2 + (\eta_t - \alpha_t)^2)\|\nabla f_t(\boldsymbol{x}_t)\|^2 - 2\eta_t \nabla f_t(\boldsymbol{x}_t)^\mathsf{T}(\boldsymbol{z}_t - \boldsymbol{x}^*)$$
$$+ \|\boldsymbol{x}_t - \boldsymbol{z}_t\|^2 + 2(\eta_t - \alpha_t)\nabla f_t(\boldsymbol{x}_t)^\mathsf{T}(\boldsymbol{x}_t - \boldsymbol{z}_t). \tag{5}$$

Expanding $-2\eta_t \nabla f_t(\boldsymbol{x}_t)^\mathsf{T}(\boldsymbol{z}_t - \boldsymbol{x}^*)$ as $-2\eta_t \nabla f_t(\boldsymbol{x}_t)^\mathsf{T}(\boldsymbol{z}_t + \boldsymbol{x}_t - \boldsymbol{x}_t - \boldsymbol{x}^*)$ and noting that

$$-\nabla f_t(\boldsymbol{x}_t)^\mathsf{T}(\boldsymbol{x}_t - \boldsymbol{x}^*) \leq f_t(\boldsymbol{x}^*) - f_t(\boldsymbol{x}_t) - \frac{1}{2L}\|\nabla f_t(\boldsymbol{x}_t)\|^2,$$

from Corollary A.8.1, it follows that

$$-2\eta_t \nabla f_t(\boldsymbol{x}_t)^\mathsf{T}(\boldsymbol{z}_t - \boldsymbol{x}^*) \leq 2\eta_t \nabla f_t(\boldsymbol{x}_t)^\mathsf{T}(\boldsymbol{x}_t - \boldsymbol{z}_t) + 2\eta_t(f(\boldsymbol{x}^*) - f(\boldsymbol{x}_t)) - \frac{\eta_t}{L_t}\|\nabla f_t(\boldsymbol{x}_t)\|^2. \tag{6}$$

Applying the bound in Equation (6) to Equation (5) we have

$$V_{t+1} \leq \|\boldsymbol{z}_t - \boldsymbol{x}^*\|^2 + \left(\eta_t^2 + (\eta_t - \alpha_t)^2 - \frac{\eta_t}{L_t}\right)\|\nabla f_t(\boldsymbol{x}_t)\|^2 + 2\eta_t(f(\boldsymbol{x}^*) - f(\boldsymbol{x}_t))$$
$$+ \|\boldsymbol{x}_t - \boldsymbol{z}_t\|^2 + 2(2\eta_t - \alpha_t)\nabla f_t(\boldsymbol{x}_t)^\mathsf{T}(\boldsymbol{x}_t - \boldsymbol{z}_t). \tag{7}$$

Now turning our attention to $V_t$ and rewriting the second component to be in terms of $\boldsymbol{x}_t - \boldsymbol{z}_t$ we have

$$V_t = \|\boldsymbol{z}_t - \boldsymbol{x}^*\|^2 + \bar{\beta}_t^{-2}\|\boldsymbol{x}_t - \boldsymbol{z}_t\|^2 \tag{8}$$

where we have used the fact that $\boldsymbol{x}_t - \boldsymbol{z}_t = (1 - \beta_t)(\boldsymbol{y}_t - \boldsymbol{z}_t)$ from Equation (1a). Subtracting (8) from (7) it follows that

$$
\begin{aligned}
\Delta V_{t+1} \leq & \left( \eta_t^2 + (\eta_t - \alpha_t)^2 - \frac{\eta_t}{L_t} \right) \left\| \nabla f_t(\boldsymbol{x}_t) \right\|^2 + 2\eta_t (f_t(\boldsymbol{x}^*) - f_t(\boldsymbol{x}_t)) \\
& + \left( 1 - \bar{\beta}_t^{-2} \right) \|\boldsymbol{x}_t - \boldsymbol{z}_t\|^2 + 2(2\eta_t - \alpha_t) \nabla f_t(\boldsymbol{x}_t)^{\mathsf{T}} (\boldsymbol{x}_t - \boldsymbol{z}_t).
\end{aligned}
\tag{9}
$$

We pause to define some variables

$$
\begin{aligned}
\boldsymbol{a}_t &:= \frac{1}{N_t} \nabla f_t(\boldsymbol{x}_t) & c_1 &:= \gamma_t^2 + (\gamma_t - \mu_t)^2 - \gamma_t & c_3 &:= 1 - \bar{\beta}_t^{-2} \\
\boldsymbol{b}_t &:= \boldsymbol{x}_t - \boldsymbol{z}_t & c_2 &:= 2(2\gamma_t - \mu_t) & c_4 &:= 2\frac{\gamma_t}{N_t}(f_t(\boldsymbol{x}^*) - f_t(\boldsymbol{x}_t))
\end{aligned}
$$

Using the above definitions and the fact that $N_t \geq L_t$ (Assumption 3) it follows that

$$
\Delta V_{t+1} \leq c_1 \|\boldsymbol{a}_t\|^2 + c_2 \boldsymbol{a}_t^{\mathsf{T}} \boldsymbol{b}_t + c_3 \|\boldsymbol{b}_t\|^2 + c_4.
$$

Completing the square we have

$$
\Delta V_{t+1} \leq c_1 \left\| \boldsymbol{a}_t + \frac{c_2}{2c_1} \boldsymbol{b}_t \right\|^2 + \left( c_3 - \frac{c_2^2}{4c_1} \right) \|\boldsymbol{b}_t\|^2 + c_4.
\tag{10}
$$

In order for $\Delta V_{t+1} \leq 0$ we need $c_1, c_4$ and $c_2^2 - 4c_1 c_3$ to be less than or equal to zero (which in turn implies that $c_3$ is less than zero as well). The coefficient $c_4$ is always less than zero because of the optimally of $\boldsymbol{x}^*$. The coefficients $c_1, c_3$ as well as the expression $c_2^2 - 4c_1 c_3$ are all less than zero given the conditions outlined in (3). It thus follows that

$$
\Delta V_{t+1} \leq 2\frac{\gamma_t}{N_t}(f_t(\boldsymbol{x}^*) - f_t(\boldsymbol{x}_t)) < 0 \ \forall \boldsymbol{x}_t \neq \boldsymbol{x}^*.
\tag{11}
$$

It then follows that $(\boldsymbol{z}_t - \boldsymbol{x}^*)$ and $(\boldsymbol{y}_t - \boldsymbol{z}_t)$ are bounded by the definition of $V_t$ in (4). With $\boldsymbol{x}^*$ fixed and clearly bounded it then immediately follows that $\boldsymbol{y}_t, \boldsymbol{z}_t \in \ell_\infty$. From (1a) and the boundedness of $\beta_t$ (by construction) it then follows that $\boldsymbol{y}_t \in \ell_\infty$. From Definition 2 (smooth) and from Assumptions 1-3 it then follows that the sequence $\left\| \nabla f_t(\boldsymbol{x}_t) \right\|$ is bounded as well. Finally, given that smooth functions on compact sets are bounded we have that $f_t(\boldsymbol{x}_t)$ is bounded and therefore $[f_t(\boldsymbol{x}^*) - f_t(\boldsymbol{x}_t)] \in \ell_\infty$ as well. With that all signals in the model are bounded $\boldsymbol{x}_t, \boldsymbol{y}_t, \boldsymbol{z}_t, f_t(\boldsymbol{x}_t), \nabla f_t(\boldsymbol{x}_t) \in \ell_\infty$.

Finally to prove convergence of the cost function to its minimum we will use properties of monotonic sequences (i.e. partial sums of positive values). By Assumption 1 $L_t$ is bounded and then by Assumption 3 we choose an $N_t$ that is bounded as well. Therefore, there exists a positive $\bar{N}$ such that $\bar{N} \geq N_t$ for all $t$. Finally, recall from (3c) that $\gamma_t \in [\epsilon/2, 1/2]$. Using these bounds for $N_t$ and $\gamma_t$, and Equation (11) it follows that

$$
\frac{\bar{N}}{\epsilon} \Delta V_{t+1} \leq f_t(\boldsymbol{x}^*) - f_t(\boldsymbol{x}_t).
\tag{12}
$$

Summing both sides, multiplying by $-1$ and recalling that $V_t$ is a non-increasing positive sequence, it follows that

$$
a_\tau := \sum_{t=0}^{\tau} [f_t(\boldsymbol{x}_t) - f_t(\boldsymbol{x}^*)] \leq -\frac{\bar{N}}{\epsilon} \sum_{t=0}^{\tau} \Delta V_{t+1} = \frac{\bar{N}}{\epsilon}[V_0 - V_{\tau+1}] \leq \frac{\bar{N}}{\epsilon} V_0 < \infty
\tag{13}
$$

With $a_\tau$ a bounded monotonic sequence it follows that $\lim_{\tau \to \infty} a_\tau$ exists (Rudin, 1976, Theorem 3.14) and therefore $\lim_{t \to \infty} \left[ f_t(\boldsymbol{x}_t) - f_t(\boldsymbol{x}^*) \right] = 0$. ∎

**Remark 2.** Note that we did not begin by providing successfully more refined bounds for $f_t(\boldsymbol{x}_t) - f_t(\boldsymbol{x}^*)$. Instead we began with a Lyapunov candidate that explicitly bounds the states of the optimizer (4), and then by summing the bound on the time difference of the Lyapunov candidate we achieve the desired result in (13). For the interested reader this approach also extends to continuous time settings where the last step

(under appropriate conditions) invokes Barbălat's Lemma (Farkas & Wegner, 2016). When $f_t = f$ is not time-varying it is commonly incorporated into the Lyapunov candidate directly (Liu et al., 2020, Equation(5)); (Ahn & Sra, 2022, Equation (2.3)). When $f_t$ is allowed to be time varying, however, it may not satisfy all the conditions to be a Lyapunov function. Namely, a Lyapunov function has to be lowerbounded by a time-invariant non-decrescent function of the parameter you want to show is bounded. For instance if you have a time varying Lyapunov candidate $V_t(\boldsymbol{x})$ you will need to show that there exists a non-decrescent function $\alpha$ where $\alpha(0) = 0$ and $0 < \alpha(\|\boldsymbol{x}\|) \leq V_t(\boldsymbol{x})$ for all $\boldsymbol{x} \neq \boldsymbol{0}$ (Kalman & Bertram, 1960, Theorem 1). With the cost function $f_t$ now explicitly varying over time the above condition can easily be violated.

### 3.2 A more general proof for smooth convex

**Theorem 1.** *Let $f_t$ satisfy Assumptions 1-3. For the HT defined in Equation (1) with arbitrary initial conditions $\{\boldsymbol{x}_0, \boldsymbol{y}_0, \boldsymbol{z}_0\}$, $N_t \geq L_t$ ($f_t$ is $L_t$ smooth by assumption) and with the algorithm hyper-parameters $\{\gamma_t, \mu_t, \beta_t\}$ along with the analysis parameters $\lambda_t \in [0, 1-\epsilon], \xi_t \in [\epsilon, \infty), \epsilon > 0$ chosen such that*

$$c_5 < 0, \quad and \quad c_7 - \frac{c_6^2}{4c_5} \leq 0 \tag{14}$$

*where*

$$c_5 := \gamma_t^2 + \xi_{t+1}(\gamma_t - \mu_t)^2 - (1 + \lambda_t)\gamma_t \qquad c_6 := 2[\xi_{t+1}(\gamma_t - \mu_t) + \gamma_t] \tag{15a}$$

$$c_7 := \xi_{t+1} - \xi_t \bar{\beta}_t^{-2} \tag{15b}$$

*then*

$$W_t = \|\boldsymbol{z}_t - \boldsymbol{x}^*\|^2 + \xi_t \|\boldsymbol{y}_t - \boldsymbol{z}_t\|^2 \tag{16}$$

*is a Lyapunov candidate and it follows that $W_t \in \ell_\infty$. Furthermore, $\lim_{t\to\infty} \left[f_t(\boldsymbol{x}_{t+1}) - f_t(\boldsymbol{x}^*)\right] = 0$.*

*Proof.* We begin by expanding the time step ahead Lyapunov candidate $W_{t+1} = \|\boldsymbol{z}_{t+1} - \boldsymbol{x}^*\|^2 + \xi_{t+1}\|\boldsymbol{y}_{t+1} - \boldsymbol{z}_{t+1}\|$ by substituting the update for $\boldsymbol{y}_{t+1}$ from (1b) and $\boldsymbol{z}_{t+1}$ from (1c) and noting that $\boldsymbol{x}_t - \boldsymbol{z}_t = \bar{\beta}_t(\boldsymbol{y}_t - \boldsymbol{z}_t)$ from (1a) where $\bar{\beta}_t = 1 - \beta_t$, as defined in (2), we get the following

$$W_{t+1} = \|\boldsymbol{z}_t - \boldsymbol{x}^*\|^2 + (\eta_t^2 + \xi_{t+1}(\eta_t - \alpha_t)^2)\|\nabla f_t(\boldsymbol{x}_t)\|^2 - 2\eta_t \nabla f_t(\boldsymbol{x}_t)^\mathsf{T}(\boldsymbol{z}_t - \boldsymbol{x}^*)$$
$$+ \xi_{t+1}\|\boldsymbol{x}_t - \boldsymbol{z}_t\|^2 + \xi_{t+1}2(\eta_t - \alpha_t)\nabla f_t(\boldsymbol{x}_t)^\mathsf{T}(\boldsymbol{x}_t - \boldsymbol{z}_t). \tag{17}$$

Expanding $-2\eta_t \nabla f_t(\boldsymbol{x}_t)^\mathsf{T}(\boldsymbol{z}_t - \boldsymbol{x}^*)$ as $-2\eta_t \nabla f_t(\boldsymbol{x}_t)^\mathsf{T}(\boldsymbol{z}_t + \boldsymbol{x}_t - \boldsymbol{x}_t - \boldsymbol{x}^*)$ and noting that

$$-\nabla f_t(\boldsymbol{x}_t)^\mathsf{T}(\boldsymbol{x}_t - \boldsymbol{x}^*) \leq (1 - \lambda_t)[f_t(\boldsymbol{x}^*) - f_t(\boldsymbol{x}_t)] - \frac{1 + \lambda_t}{2L}\|\nabla f_t(\boldsymbol{x}_t)\|^2,$$

from Lemma A.9, it follows that

$$-2\eta_t \nabla f_t(\boldsymbol{x}_t)^\mathsf{T}(\boldsymbol{z}_t - \boldsymbol{x}^*) \leq 2\eta_t \nabla f_t(\boldsymbol{x}_t)^\mathsf{T}(\boldsymbol{x}_t - \boldsymbol{z}_t) + 2(1 - \lambda_t)\eta_t(f_t(\boldsymbol{x}^*) - f_t(\boldsymbol{x}_t))$$
$$- \frac{(1 + \lambda_t)\eta_t}{L_t}\|\nabla f_t(\boldsymbol{x}_t)\|^2. \tag{18}$$

Substitution of Equation (18) into Equation (17)

$$W_{t+1} \leq \left(\eta_t^2 + \xi_{t+1}(\eta_t - \alpha_t)^2 - \frac{(1 + \lambda_t)\eta_t}{L_t}\right)\|\nabla f_t(\boldsymbol{x}_t)\|^2 + 2(1 - \lambda_t)\eta_t(f_t(\boldsymbol{x}^*) - f_t(\boldsymbol{x}_t))$$
$$+ \|\boldsymbol{z}_t - \boldsymbol{x}^*\|^2 + \xi_{t+1}\|\boldsymbol{x}_t - \boldsymbol{z}_t\|^2 + [\xi_{t+1}2(\eta_t - \alpha_t) + 2\eta_t]\nabla f_t(\boldsymbol{x}_t)^\mathsf{T}(\boldsymbol{x}_t - \boldsymbol{z}_t). \tag{19}$$

Turning our attention to $W_t$ and rewriting the second component to be in terms of $\boldsymbol{x}_t - \boldsymbol{z}_t$ we have

$$W_t = \|\boldsymbol{z}_t - \boldsymbol{x}^*\|^2 + \xi_t \bar{\beta}_t^{-2}\|\boldsymbol{x}_t - \boldsymbol{z}_t\|^2 \tag{20}$$

where we have used the fact that $\boldsymbol{x}_t - \boldsymbol{z}_t = (1 - \beta_t)(\boldsymbol{y}_t - \boldsymbol{z}_t)$ from Equation (1a). Substracting (20) from (19) with $\Delta W_{t+1} = W_{t+1} - W_t$ it follows that

$$\Delta W_{t+1} \leq \left( \eta_t^2 + \xi_{t+1}(\eta_t - \alpha_t)^2 - \frac{(1+\lambda_t)\eta_t}{L_t} \right) \|\nabla f_t(\boldsymbol{x}_t)\|^2 + 2(1-\lambda_t)\eta_t(f_t(\boldsymbol{x}^*) - f_t(\boldsymbol{x}_t))$$
$$+ \left( \xi_{t+1} - \xi_t \bar{\beta}_t^{-2} \right) \|\boldsymbol{x}_t - \boldsymbol{z}_t\|^2 + [\xi_{t+1} 2(\eta_t - \alpha_t) + 2\eta_t] \nabla f_t(\boldsymbol{x}_t)^\mathsf{T} (\boldsymbol{x}_t - \boldsymbol{z}_t).$$

Recall the definitions $\boldsymbol{a}_t := \frac{1}{N_t} \nabla f_t(\boldsymbol{x}_t)$ and $\boldsymbol{b}_t := \boldsymbol{x}_t - \boldsymbol{z}_t$ with $c_5, c_6, c_7$ defined in (15) along with

$$c_8 := 2 \frac{(1-\lambda_t)\gamma_t}{N_t} (f_t(\boldsymbol{x}^*) - f_t(\boldsymbol{x}_t))$$

and the fact that $N_t \geq L_t$, then it follows that $\Delta W_{t+1} \leq c_5 \|\boldsymbol{a}_t\|^2 + c_6 \boldsymbol{a}_t^\mathsf{T} \boldsymbol{b}_t + c_7 \|\boldsymbol{b}_t\|^2 + c_8$. Completing the square we have $\Delta W_{t+1} \leq c_5 \left\| \boldsymbol{a}_t + \frac{c_6}{2c_5} \boldsymbol{b}_t \right\|^2 + \left( c_7 - \frac{c_6^2}{4c_5} \right) \|\boldsymbol{b}_t\|^2 + c_8$. By the conditions in (14) it follows that

$$\Delta W_{t+1} \leq 2 \frac{(1-\lambda_t)\gamma_t}{N_t} (f_t(\boldsymbol{x}^*) - f_t(\boldsymbol{x}_t)) < 0 \quad \forall \boldsymbol{x}_t \neq \boldsymbol{x}^*.$$

Following the same steps as outlined at the end of the proof of Proposition 1 it follows that $a_\tau := \sum_{t=0}^\tau [f_t(\boldsymbol{x}_t) - f_t(\boldsymbol{x}^*)]$ is a bounded monotonic sequence and therefore $\lim_{t \to \infty} [f_t(\boldsymbol{x}_t) - f_t(\boldsymbol{x}^*)] = 0$. ∎

**Remark 3.** With a slightly modified Lyapunov candidate (compare (16) to (4)) we get the most general bound we are aware of. Note however that the learning rate parameter $\gamma_t$ must be less than 2 for stability to hold, see (14) and (15a). We raise this point because when the function $f_t$ is fixed for all time ($f_t = f$: a constant) Nesterov's accelerated method requires $\gamma_t$ to increase with time, $t$, in an unbounded fashion (i.e. $\gamma_t \asymp t$). We discuss this further in §4.2 – it is likely impossible to have a stable accelerated method like Nesterov for arbitrary time varying functions. We follow up the main theorem with a corollary that has much simpler conditions with some accompanying simulations to demonstrate the non-vacuous nature of our bounds.

**Corollary 1.1.** *Let $f_t$ satisfy Assumptions 1-3. For the HT defined in Equation (1) with arbitrary initial conditions $\{\boldsymbol{x}_0, \boldsymbol{y}_0, \boldsymbol{z}_0\}$, $N_t \geq L_t$ ($f_t$ is $L_t$ smooth by assumption)*

$$\gamma_t \in [1, \ 1.5] \tag{21a}$$
$$\mu_t = 1 \tag{21b}$$
$$\beta_t = 1/\gamma_t \tag{21c}$$

*and where the free design parameters (that only appear in the analysis) are chosen as $\lambda_t = 1$ and $\xi_t = 1$, then all states and gradients are uniformly bounded. If we replace the above bounds with the slightly more stringent cases where the free design parameter $\lambda_t < 1$ which in turn will require $\gamma_t < 1.5$ then we also have $\lim_{t \to \infty} [f_t(\boldsymbol{x}_t) - f_t(\boldsymbol{x}^*)] = 0$. A visualization of the stability conditions in (14) for the parameters in (21) are shown in Figure 1.*

We have an entire section dedicated to simulations but we didn't want to wait until the very end to demonstrate the non-vacuous nature of our results. The simulation in Figure 1(B) demonstrate that with $\gamma_t = 1$ the HT coincides with vanilla GD. As $\gamma_t$ increases beyond 1 (with $\beta_t = 1/\gamma_t$, $\mu_t = 1$) the HT (empirically) demonstrates acceleration (monochrome black/white solid lines) with the purple dashed line coinciding with $\gamma_t = 1.5$ matching our stability analysis in Corollary 1.1. In the simulation we also changed the optimal point halfway through so as to have another window of transients where the algorithms have to converge to the new optimal value. As was previously stated in Remark 1 our stability results (bounded trajectories for arbitrary initial conditions) hold even for optimal set points that vary over time. We will have more to say about the long term behavior of the HT in Section 5.

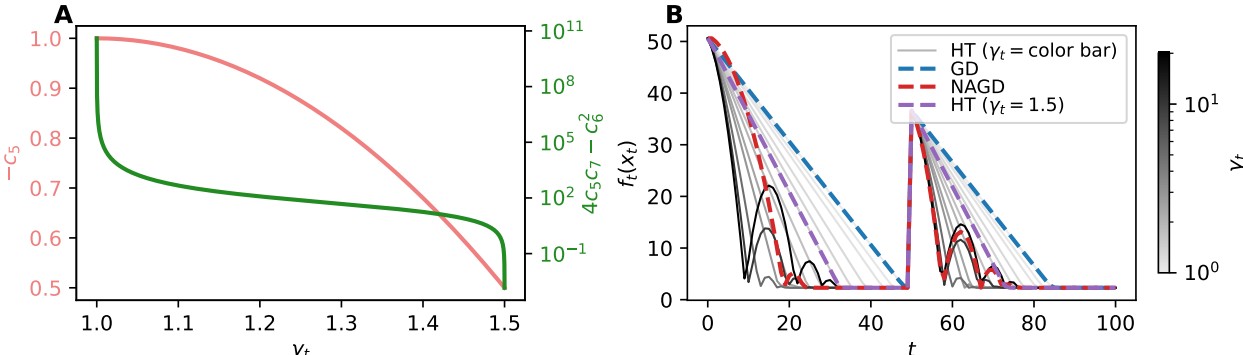

**Figure 1: The stability conditions outlined in Corollary 1.1 are nonvacuous. (A)** Visualization of the sufficient conditions outlined in Equation (14) under the parameter settings in Equation (21). There is an asymptote at $\gamma_t = 1.5$ for the value of $4c_5c_7 - c_6^2$ (green line) **(B)** Algorithms are optimizing the function $f_t(x) = \log(a_t \exp(-b_t(x - c_t)) + a_t \exp(b_t(x - c_t)))$ where $a_t = 1$ and $b_t = 7$. For $t < 50$ $c_t = 0$ and for $t > 50$ $c_t = 5$. This results in an optimization problem where the optimal solution $x^*$ changes from 0 to 5 at $t = 50$. Vanilla GD is shown in the dashed blue line with Nesterov Accelerated Gradient Descent (NAGD) in dashed red. For the HT with $\gamma_t$ varied from 1 to 10 see the solid monochrome (shades of grey) lines. With $\gamma_t = 1$ the optimizer coincides with GD and as $\gamma_t$ is increased the HT begins to exhibit acceleration more closely resembling NAGD (light grey to black solid line). HT with $\gamma_t = 1.5$ shown in dashed purple. Full details provided in Appendix B.

### 3.3 The smooth and strongly convex setting

We close by showing exponential convergence for time varying strongly convex functions.

**Proposition 2.** *Let $f_t$ satisfy Assumptions 1-3 as well as the additional assumption that for each $t$, $f_t$ is $\sigma_t$−strongly convex. For the HT defined in Equation (1) with arbitrary initial conditions $\{\boldsymbol{x}_0, \boldsymbol{y}_0, \boldsymbol{z}_0\}$, $N_t \geq L_t$ ($f_t$ is $L_t$ smooth by assumption), and the same three conditions from Equation (3) in Proposition 1 satisfied $\{\beta_t \in [0, 1], \mu_t \in [\epsilon, 1], \epsilon > 0, \gamma_t = \frac{\mu_t}{2}\}$. Then, $V_t = \|\boldsymbol{z}_t - \boldsymbol{x}^*\|^2 + \|\boldsymbol{y}_t - \boldsymbol{z}_t\|^2$ originally defined in (4) is a Lyapunov function which exponentially converges to zero at rate*

$$V_{t+1} = (1 - \omega_t) V_t \tag{22}$$

*where*

$$\omega_t := \min\left\{\left[(1 - \nu_t)(1 - \bar{\beta}_t^2)\right], \left[\frac{\rho_t \nu_t(1 - \bar{\beta}_t^2)}{\rho_t \bar{\beta}_t^2 + \nu_t(1 - \bar{\beta}_t^2)}\right]\right\}, \tag{23a}$$

$$\rho_t := \frac{\mu_t}{2}\frac{\sigma_t}{N_t}, \quad and \quad \nu_t \in [\epsilon, 1 - \epsilon]. \tag{23b}$$

*Proof.* In order to demonstrate exponential convergence with the Lyapunov function it suffices to show $V_{t+1} = g_t V_t$ for some $g_t \in [-1 + \epsilon, 1 - \epsilon]$. We will begin the proof by starting where we left off with the bound (10) in Proposition 1, expanding the coefficients and substituting $\gamma_t = \mu_t/2$ which results in

$$\Delta V_{t+1} \leq \frac{\mu_t}{N_t^2}(\mu_t - 1)\left\|\nabla f_t(\boldsymbol{x}_t)\right\|^2 + \left(1 - \bar{\beta}_t^{-2}\right)\|\boldsymbol{x}_t - \boldsymbol{z}_t\|^2 + \frac{\mu_t}{N_t}(f_t(\boldsymbol{x}^*) - f_t(\boldsymbol{x}_t)) \tag{24a}$$

$$\leq \left(1 - \bar{\beta}_t^{-2}\right)\|\boldsymbol{x}_t - \boldsymbol{z}_t\|^2 + \frac{\mu_t}{N_t}(f_t(\boldsymbol{x}^*) - f_t(\boldsymbol{x}_t)) \tag{24b}$$

where we dropped the first term in (24a) for the second inequality in (24b) because we typically want the learning rate $\mu_t \in [\epsilon, 1]$ to be as large as possible and the first expression in (24a) approaches 0 as $\mu_t$ tends to 1. Next we will use the strong convexity assumption along with the distance to the minimizer bound provided in Lemma A.7 for $f_t(\boldsymbol{x}^*) - f_t(\boldsymbol{x}_t)$ and we get

$$\Delta V_{t+1} \leq \left(1 - \bar{\beta}_t^{-2}\right)\|\boldsymbol{x}_t - \boldsymbol{z}_t\|^2 - \rho_t\|\boldsymbol{x}_t - \boldsymbol{x}^*\|^2 \tag{25}$$

where we recall the definition of $\rho_t$ from (23). The next step is to obtain bounds in terms $\|z_t - x^*\|$ and $\|y_t - z_t\|$ with the hopes that the upper-bound can be expressed in terms $V_t$ thus allowing for some kind of bound of the form $V_{t+1} = g_t V_t$ as we previously stated. We accomplish this by adding and subtracting $z_t$ within the norm $\|x_t - x^*\|$ and then using the fact that $x_t - z_t = \bar{\beta}_t(y_t - z_t)$ from (1a) we obtain

$$\Delta V_{t+1} \leq (\bar{\beta}_t^2 - 1)\|y_t - z_t\|^2 - \rho_t\left(\bar{\beta}_t^2\|y_t - z_t\|^2 + 2\bar{\beta}_t(y_t - z_t)^\intercal(z_t - x^*) + \|z_t - x^*\|^2\right). \tag{26}$$

As in our prior analysis we define intermediate vectors and scalars to simplify the expressions with

$$c_t := y_t - z_t \qquad d_t := z_t - x^* \qquad \psi_t^2 := \bar{\beta}_t^2 - 1 \quad \text{(note that } \psi_t \leq 0) \tag{27}$$

and rewrite the inequality in (26) giving us the more compact inequality

$$\Delta V_{t+1} \leq \psi_t^2\|c_t\|^2 - \rho_t\left(\bar{\beta}_t^2\|c_t\|^2 + 2\bar{\beta}_t c_t^\intercal d_t + \|d_t\|^2\right).$$

We want to have an expression that is negative semidefinite in terms of $\|c\|_t$ and $\|d\|_t$ which we have for $\|c\|_t$, but not yet for $\|d\|_t$. To achieve this we borrow $\nu_t \in [\epsilon, 1 - \epsilon]$ from the expression $\psi_t^2\|c_t\|$ and incorporate it into the large expression in the parenthesis giving us

$$\Delta V_{t+1} \leq (1 - \nu_t)\psi_t^2\|c_t\|^2 + \left[\left(-\rho_t\bar{\beta}_t^2 + \nu_t\psi_t^2\right)\|c_t\|^2 - 2\rho_t\bar{\beta}_t c_t^\intercal d_t - \rho_t\|d_t\|^2\right]. \tag{28}$$

Completing the square for the component in the brackets we have

$$\left(-\rho_t\bar{\beta}_t^2 + \nu_t\psi_t^2\right)\left\|c_t + \frac{-2\rho_t\bar{\beta}_t}{2\left(-\rho_t\bar{\beta}_t^2 + \nu_t\psi_t^2\right)}d_t\right\|^2 + \left(-\rho_t - \frac{\rho_t^2\bar{\beta}_t^2}{\left(-\rho_t\bar{\beta}_t^2 + \nu_t\psi_t^2\right)}\right)\|d_t\|^2. \tag{29}$$

Taking the second component from (29) that is solely a function of $\|d_t\|^2$ and using that to bound the expression in the brackets of Equation (28) we obtain

$$\Delta V_{t+1} \leq -(1 - \nu_t)(1 - \bar{\beta}_t^2)\|c_t\|^2 - \frac{\rho_t\nu_t(1 - \bar{\beta}_t^2)}{\rho_t\bar{\beta}_t^2 + \nu_t(1 - \bar{\beta}_t^2)}\|d_t\|^2 \tag{30}$$

Finally, we recognize from the definition of $c_t$ and $d_t$ that $V_t = \|c_t\|^2 + \|d_t\|^2$, and from Equation (30) with a few terms rearranged we arrive at the bound provided in Equation (22). ∎

### 3.4 Time varying optimal points

In Remark 1 we eluded to the fact that while we assume a fixed optimal point $x^*$, for the purposes of proving asymptotic convergence results, we can actually allow for time varying $x_t^*$ and we still remain bounded. Specifically, for our setting one can prove that for any piecewise constant $x_t^*$ with only a finite number of changes the HT remains bounded with $f_t(x_t)$ converging to each new optimal point after it is changed. This holds because in our proof setting we make no assumptions about the initial conditions of the parameters or any other a priori assumptions about their boundedness. We can make this a little more concrete with the following thought experiment.

Consider our problem setting with arbitrary initial conditions $x_0, y_0, z_0$ and an optimal $x_{[1]}^*$ to begin with. Our proofs say that we are stable and $f_t(x_t)$ will asymptotically converge to $f_t(x_{[1]}^*)$. Now at some time $T$ we get a new optimal $x_{[2]}^*$ (with the subscript [2]). Importantly we do nothing to the algorithm and it just continues to run as originally defined. For analysis we now say $x_T, y_T, z_T$ are our new initial conditions. Recall that our proof explicitly states that we are stable for arbitrary initial conditions and so we will still be bounded for all time with the new $x_{[2]}^*$ and $f_t(x_t)$ will now asymptotically converge to the new $f_t(x_{[2]}^*)$ as well. Using this line of reasoning the following meta-theorem holds.

**Theorem 2.** *Let $f_t$ satisfy Assumptions 1 and 3 where there exists a piecewise constant $\boldsymbol{x}_t^*$ that changes in value only a finite number of times. For the HT defined in Equation (1) with arbitrary initial conditions $\{\boldsymbol{x}_0, \boldsymbol{y}_0, \boldsymbol{z}_0\}$, $N_t \geq L_t$ ($f_t$ is $L_t$ smooth by assumption), and the conditions on the hyper-parameters and further assumptions as outlined in either Propositions 1 and 2 or Theorem 1, the parameters of the HT remain bounded for all time. Furthermore after each successive change in the optimal point, $f_t(\boldsymbol{x}_t)$ will begin to asymptotically converge (or $\boldsymbol{x}_t$ will begin to exponentially converge in the case of strongly convex) to each new optimal function value (to each new optimal point).*

One can go further and analyze continuously varying $\boldsymbol{x}_t^*$, for instance if $\boldsymbol{x}_t^*$ is "slowly varying" (the quantity $\boldsymbol{x}_{t+1}^* - \boldsymbol{x}_t^*$ is uniformly bounded) and the style of analysis presented here can be adapted to this setting as well. However, some kind of robustness modification will be needed (e.g. the use of the projection operator or leaky integration, which is called the "sigma" modification in the adaptive control literature Tsakalis & Ioannou (1987)). One can no longer prove convergence to the optimal functions or optimal point themselves, but instead, compact regions around them (Narendra & Annaswamy, 2012, §8.6). We note that these notions of stability never enters one's mind when studying gradient algorithms with tools like regret analysis because you a priori assume all the parameters and gradients are uniformly bounded before you even start the analysis, see §4.3.

# 4 Connections to existing literature

## 4.1 High-order tuner

The HT in (1) was directly inspired by Gaudio et al. (2021) which tried to connect stability proof techniques for second order gradient flow methods (continuous time) to second order gradient descent methods (discrete time). The second order gradient flow algorithm under study in Gaudio et al. (2021) was $\ddot{\boldsymbol{x}}(t) + \beta\dot{\boldsymbol{x}}(t) = -\frac{\gamma\beta}{N_t}\nabla f_t(\boldsymbol{x}(t))$ which the authors recognized could be reparameterized as

$$\dot{\boldsymbol{z}}(t) = -\frac{\gamma}{N_t}\nabla f_t(\boldsymbol{x}(t)) \tag{31a}$$

$$\dot{\boldsymbol{x}}(t) = -\beta(\boldsymbol{x}(t) - \boldsymbol{z}(t)) \tag{31b}$$

where $N_t$ varies over time and upper bounds the smoothness of $f_t$ at each $t$. Reparameteriztion to the form in Equation (31) was essential for discovery of the Lyapunov candidate

$$V(t) = \frac{1}{\gamma}\left(\left\|\boldsymbol{z}(t) - \boldsymbol{x}^*\right\|^2 + \left\|\boldsymbol{z}(t) - \boldsymbol{x}(t)\right\|^2\right). \tag{32}$$

What was also shown in Gaudio et al. (2021) was that the discretization of Equation (31) to

$$\boldsymbol{x}_t = \beta\boldsymbol{z}_t + (1 - \beta)\boldsymbol{y}_t \tag{33a}$$

$$\boldsymbol{y}_{t+1} = \boldsymbol{x}_t - \frac{\gamma\beta}{N_t}\nabla f_{t+1}(\boldsymbol{x}_t) \tag{33b}$$

$$\boldsymbol{z}_{t+1} = \boldsymbol{z}_t - \frac{\gamma}{N_t}\nabla f_t(\boldsymbol{x}_t) \tag{33c}$$

was also stable and could use the same Lyapunov function (but in discrete time $V_t = \frac{1}{\gamma}\|\boldsymbol{z}_t - \boldsymbol{x}^*\|^2 + \frac{1}{\gamma}\|\boldsymbol{z}_t - \boldsymbol{x}_t\|^2$). For quadratic costs and the HT in (33) they established stability with the following constraints on their free design parameters from (Gaudio et al., 2021, Theorem 4)

$$\beta \in (0, 1) \quad \gamma \in \left(0, \frac{\beta(2-\beta)}{16+\beta^2}\right] \tag{34}$$

and for quadratic costs that are $\sigma$-strongly convex a similar but even more conservative bound was established (Gaudio et al., 2021, Theorem 5). Moreu & Annaswamy (2022) used the same algorithm and analysis to provide similarly conservative bounds for more general convex functions (Moreu & Annaswamy, 2022, Theorems 1,2). These overly conservative constraints do not allow $\gamma_t$ to exceed 1 (or even approach 1) which is necessary to achieve a performance improvement over vanilla GD (Corollary 1 and Figure 1).

There are several differences provided in this work compared to the prior art (Gaudio et al., 2021; Moreu & Annaswamy, 2022). First we note that in our analysis, the free design parameters $\gamma_t$ and $\beta_t$ (Equations (1) and (2)) are allowed to be time varying in comparison to $\gamma, \beta$ in (33). We also added an extra degree of freedom $(\alpha_t, \mu_t)$ in the gradient step size for the update $\boldsymbol{y}_{t+1}$ (compare (33b) to (1b) and (2)). Our analysis (even for the simplified constraints in Proposition 1 and Corollary 1.1) allow more flexibility in the choice of $\gamma_t$ and $\beta_t$ (compare Equations (3), (14), (21) to (34)). We also used a different Lyapunov candidate throughout, not scaling by $\gamma_t$ and instead of using $\|\boldsymbol{z}_t - \boldsymbol{x}_t\|$ we used $\|\boldsymbol{z}_t - \boldsymbol{y}_t\|$. In their work both of these differences are moot because $\gamma_t, \beta_t$ are in fact treated as constants so we can easily map between the Lypaunov functions. In our work they are not constants and these subtle changes to the Lyapunov candidate have non-trivial impacts on the analysis. We also simplified the discrete HT, not requiring one to pass the state $\boldsymbol{x}_t$ through gradients for $f$ at two different time points ($\nabla f_{t+1}(\boldsymbol{x}_t)$ and $\nabla f_t(\boldsymbol{x}_t)$ in (33) as compared to just $\nabla f_t(\boldsymbol{x}_t)$ in (1)). Finally, for the strongly convex setting we establish exponential convergence of the Lyapunov function to zero, where as prior art established exponential convergence to a compact set (compare Proposition 2 to (Gaudio et al., 2021, Theorem 5) and (Moreu & Annaswamy, 2022, Theorem 2)).

## 4.2 Accelerated methods (classical)

While the structure of our second order method is nearly identical to the setting one has when studying accelerated methods, there are some distinct differences. The first and foremost is that in our setting, we are assuming that the objective function is changing over time, our cost functions are subscripted with $t$ explicitly denoting their time variation, Assumption 1. If we set $f_t := f$ and set our hyper-parameters as

$$\mu_t = 1 \qquad \gamma_t = \frac{t}{2} \qquad \beta_t = \frac{2}{t+1} \tag{35}$$

then we exactly recover Nesterov form II (Ahn & Sra, 2022; Nesterov, 2018). The biggest difference between these parameters and ours is that in order for us to establish stability the hyper-parameter $\gamma_t$ can not grow unbounded and in our case could reach a maximum of 2 (compare (21a) to (35)). For any stability based approach with time varying $f_t$, it does not seem possible for a component of the algorithm to have a learning rate that scales beyond $\mathcal{O}(1/N_t)$ as in the case for Nesterov. One of the issues is that the state $\boldsymbol{z}_t$ does indeed grow unbounded in Nesterov. This however is not an issue when implementing Nesterov as the parameter one cares about is actually the component of $\boldsymbol{z}_t$ that contributes to $\boldsymbol{x}_t$, which from (1a) we see is $\beta_t \boldsymbol{z}_t$. To that end we tried to analyze the equivalent dynamics with

$$\tilde{\boldsymbol{x}}_t = \tilde{\boldsymbol{z}}_t + (1 - \beta_t)\tilde{\boldsymbol{y}}_t \tag{36a}$$

$$\tilde{\boldsymbol{y}}_{t+1} = \tilde{\boldsymbol{x}}_t - \alpha_t \nabla f_t(\tilde{\boldsymbol{x}}_t) \tag{36b}$$

$$\tilde{\boldsymbol{z}}_{t+1} = \frac{\beta_{t+1}}{\beta_t}\tilde{\boldsymbol{z}}_t - \beta_{t+1}\eta_t \nabla f_t(\tilde{\boldsymbol{x}}_t) \tag{36c}$$

where $\tilde{\boldsymbol{z}}_t := \beta_t \boldsymbol{z}_t$ and the other variables are unchanged but just redefined as $\tilde{\boldsymbol{x}}_t = \boldsymbol{x}_t$ and $\tilde{\boldsymbol{y}}_t = \boldsymbol{y}_t$. So now the effective learning rate associated with the state $\tilde{z}_t$ would be $\beta_{t+1}\eta_t = \frac{t}{t+2}\frac{1}{N_t}$ which is indeed proportional to $1/N_t$, but we were unsuccessful with this approach.

Others have pointed out stability issues with Nesterov's method for smooth convex functions, namely Attia & Koren (2021). Much of the prior work has focused on different definitions of stability and constructing careful examples for a single function $f$ with desired properties. In our fully time varying setting however no such careful construction is necessary and the building blocks of the "gadget" function in Attia & Koren (2021) can be used directly as a counter example against stability for our algorithm with hyper-parameters as they are defined in (36) when applied to time varying cost functions. Furthermore, while Nesterov's method does indeed have some stability guarantees for fixed quadratic cost functions (Chen et al., 2018; Lessard et al., 2016), we conjecture that those guarantees vanish when the quadratic cost functions vary over time.

## 4.3 Online learning and stochastic gradient descent

Our general setting of a cost function changing at each iteration also naturally arises in the setting of online learning and various flavors of stochastic (or sub-gradient) descent where for a fixed cost function the data

arrives in a time varying fashion (Shalev-Shwartz, 2012; Hardt et al., 2016). We can make this explicit by assuming we have a set of data $\mathcal{D}$ and at time $t$ we use a piece of that data $\boldsymbol{D}_t \in \mathcal{D}$ to define our time varying cost function $f_t(\cdot) := f(\cdot, \boldsymbol{D}_t)$ from the fixed but data dependent cost $f$.

When addressing these problems in the online learning setting Regret Analysis (RA) is used and this requires the learning rates to asymptotically decrease over time (Shalev-Shwartz, 2012).[4] This is accomplished either through explicitly reducing the learning rates over time or implicitly via scaling the learning by the accumulated sub-gradients (e.g. Adagrad (Duchi et al., 2011) and Adam (Kingma & Ba, 2015)). Another major difference is in the initial problem setup and what assumptions are made. In the analysis of Adam for instance most signals and gradients in the algorithm are *a priori* assumed to be bounded[5] with projection operators also extensively used in RA.

There are several recent attempts to address the stability of Stochastic Gradient Descent with Momentum (SGDM) for general smooth convex functions (Liu et al., 2020; Yan et al., 2018). That work is more in line with the analysis presented here. One key difference, however, is in the choice of Lyapunov candidate. In SGDM (and in the analysis of Nesterov's accelerated method) a quantity like $\|f_t(\boldsymbol{y}_t) - f_t(\boldsymbol{x}^*)\|$ is directly bounded or included in the Lyapunov candidate, where in our work, the function $f$ does not appear in the Lyapunov candidate (compare (Liu et al., 2020, Equation (5)), (Ahn & Sra, 2022, Equation (4.10)), (Wilson et al., 2021, Equation (10)), or (Bansal & Gupta, 2019, Equation (5.5)) to Equation (4) in this work) for the reasons we previously discussed in Remark 2. It instead falls out in the difference of the Lyapunov candidate, see Equation (11). This subtle difference likely has non-trivial implications. One exciting direction forward would be to possibly combine the Lyapunov candidate in this work with the stochastic analysis by Liu et al. (2020).

## 5 Simulation experiments

With these simulations we have two goals. The first is to demonstrate what can happen to traditional vanilla GD and NAGD when they are applied to time varying cost functions. The second is to demonstrate the long term behavior of HT in comparison to methods like Adam and Adagrad, which were designed via regret analysis. For all of our experiments, including the ones we already discussed, the following function is being optimized

$$f_t(x) = \log \left( a_t \exp(-b_t(x - c_t)) + a_t \exp(b_t(x - c_t)) \right). \tag{37}$$

Throughout the experiments we are comparing the HT with hyper-parameters $\gamma_t = 1.5$, $\mu_t = 1$, and $\beta_t = 1/1.5$ (Corollary 1.1) to three different groups of methods, GD and NAGD, Time-varying Normalized (TN)-GD and TN-NAGD, and Adagrad and Adam. For completeness we list their details as well as hyper-parameter settings below. For complete details on these experiments see Section B

- GD and NAGD (Figures 1 and 2A)
    - GD : $x_{t+1} = \boldsymbol{x}_t - \frac{1}{N}\nabla f_t(\boldsymbol{x}_t)$, (normalization parameter $N$ is constant).
    - NAGD: Dynamics in (1) with parameters chosen as in (35) with $\mu_t = 1$, $\gamma_t = \frac{t}{2}$, $\beta_t = \frac{2}{t+1}$ (normalization parameter $N_t = N$ is constant).
- TN-GD and TN-NAGD (Figure 2B)
    - TN-GD : $x_{t+1} = \boldsymbol{x}_t - \frac{1}{N_t}\nabla f_t(\boldsymbol{x}_t)$, (normalization parameter $N_t$ changes over time).
    - TN-NAGD: Dynamics in (1) with parameters chosen as in (35) with $\mu_t = 1$, $\gamma_t = \frac{t}{2}$, $\beta_t = \frac{2}{t+1}$ (normalization parameter $N_t$ can now vary over time).
- Adagrad and Adam (Figure 3)
    - Adagrad: $x_{t+1} = \boldsymbol{x}_t - \alpha \frac{\nabla f_t(\boldsymbol{x}_t)}{\sqrt{\sum \nabla f_\tau(x_\tau)^2} + \epsilon}$, $\alpha$ is a constant

---

[4]Adaptive regret analysis (Hazan & Seshadhri, 2009) looks to overcome some of the drawbacks mentioned here but we are not aware of their use in the analysis of gradient methods with second order dynamics.

[5]"Assume that the function $f_t$ has bounded gradients, $\|\nabla f_t(\theta)\|_2 \leq G$, $\|\nabla f_t(\theta)\|_\infty \leq G_\infty$ for all $\theta \in R^d$ and distance between any $\theta_t$ generated by Adam is bounded, $\|\theta_n - \theta_m\|_2 \leq D$, $\|\theta_m - \theta_n\|_\infty \leq D_\infty$ for any $m, n \in \{1, ..., T\}$" – Kingma & Ba (2015, page 4, Theorem 4.1)

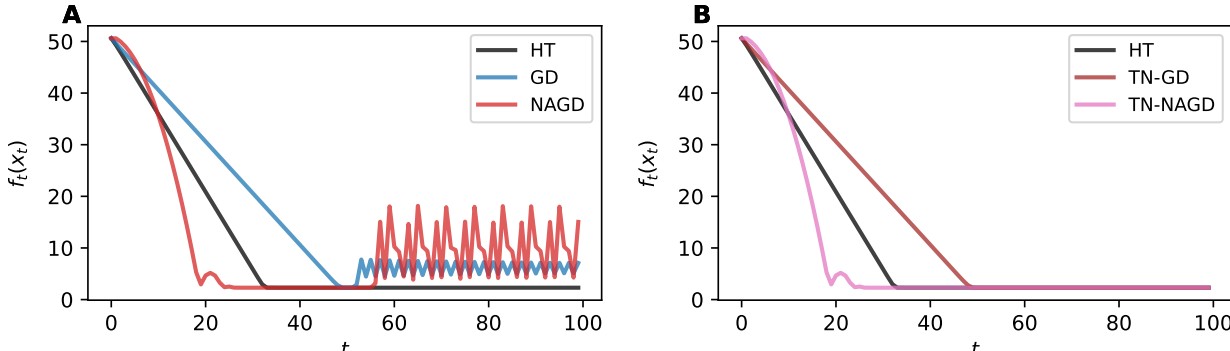

**Figure 2: An abrupt change in the smoothness of the objective function can cause instability.** At $t = 50$ the parameter $b_t$ in (37) is changed from 7 to 14. **(A)** The normalizer for the learning rate, $N$, is designed for $b_t = 7$ and when $b_t = 14$ the normalizer is too small and GD and NAGD become unstabe. The HT is provably stable with time varying $N_t$ and this parameter is adjusted to accommodate for the change in the smoothness of the objective function. **(B)** When the classic algorithms are given a time-varying normalizer they do not demonstrate the same poor performance. TN-GD is provably stable, but there is not such theory for TN-NAGD. HT is the same in both (A) and (B).

$$- \text{ Adam: } x_{t+1} = \boldsymbol{x}_t - \alpha_t \frac{\hat{m}_t}{\sqrt{\hat{v}_t} + \epsilon}, \; \alpha_t = \frac{\alpha}{\sqrt{t}}$$

In Figure 2A we are simply showing what happens with vanilla GD and NAGD when an unanticipated change occurs in the smoothness of the function that is being optimized. If the smoothness of the function becomes large relative to the normalization constant in GD and NAGD then instability can occur. There is a trivial and obvious fix to this problem. One can simply replace the fixed value of $N$ in the denominator of the learning rates of GD and NAGD with a time varying parameter $N_t$ resulting in TN-GD and TN-NAGD, see Figure 2B. This is an obvious thing one can do and for TN-GD it can actually be shown that this is still a stable algorithm. *It is not at all obvious however that one can simply adjust hyper-parameters on the fly for NAGD or any second order gradient descent algorithm for that matter.* This was one of the main motivations for this work. With the HT we can adjust $N_t$ on the fly and we are provably stable.

The HT is designed to run continuously. As previously discussed in the introduction, while the name 'online learning' might suggest learning continuously, it does not in fact mean this. We compared the HT to two prominent methods Adam and Adagrad that were designed via regret analysis. In Figure 3 the HT, Adam, and Adagrad are compared where at three different times throughout the experiment the optimal point in the objective function is changed (this is performed by changing $c_t$ in (37)). In this experiment the learning rate $\alpha$ for Adam and Adagrad was manually tuned for the best performance over the first 50 time steps of the experiment. One can see that indeed the yellow and green lines converge much faster to the minimum over the first 50 time steps. However, with each successive change in the optimal point of the objective function the performance of Adam and Adagrad degrades. This kind of behavior is expected from any algorithm designed via regret analysis as the learning rates have to asymptotically decay.

We have not yet explicitly defined regret, so lets do that here.

$$\texttt{Regret}(T) := \sum_{t=1}^{T} f_t(\boldsymbol{x}_t) - \sum_{t=1}^{T} f_t(\bar{\boldsymbol{x}}_T) \quad \text{where} \quad \bar{\boldsymbol{x}}_T = \arg\min_{\boldsymbol{x}} \sum_{t=1}^{T} f_t(\boldsymbol{x}). \tag{38}$$

With regret the accumulated cost up to time $T$ is always compared to the accumulated cost with the best fixed cost in hindsight $\bar{\boldsymbol{x}}_T$. In Figure 3(B) you can compare the pointwise in time optimal solution ($\boldsymbol{x}_t^*$, grey-dashed line) to the regret optimal solution ($\bar{\boldsymbol{x}}_T$, pink). The regret optimal solution spend much of its time near 0 because it is trying to figure out the best fixed optimal value up until that time. Inspecting Figure 3(C) we see that all the methods have average costs that lower than the average cost with the regret optimal solution. In Figure 3(D) we plotted the methods average regret over time (the values from panel (C) subtracted from the pink line) as well as the lower bound where regret is calculated for the pointwise in time optimal solution.

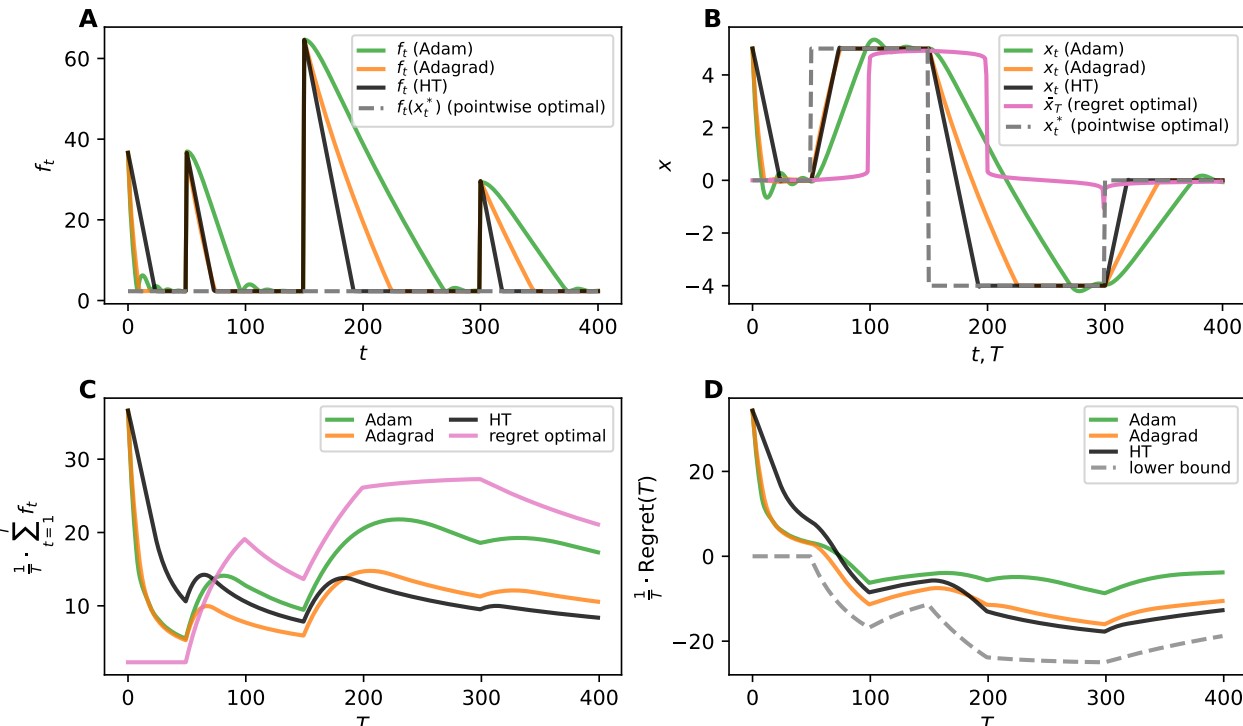

**Figure 3: Methods designed via regret analysis have degraded performance over time.** The parameter that controls the optimal solution $c_t$ in (37) is initially at 0, is changed to 5 at $t = 50$, is changed to $-4$ at $t = 150$, and then is returned to 0 at $t = 300$. **(A)** The objective function $f_t$ over time. One can see that the performance of Adam and Adagrad degrades over time. **(B)** The estimates $x_t$ from the optimizers along with the regret optimal and pointwise optimal solutions. **(C)** Average cost over time. **(D)** Average regret over time.

## 6 What does acceleration look like with time varying cost functions?

We haven't discussed acceleration with any meaningful depth yet. We will cut to the chase — ***barring extra assumptions there really is no such thing as acceleration when one allows for time varying cost functions***. You can have a time-varying cost function with a time-invariant optimal solution (Assumption 2) for which all gradient based optimization technique will have arbitrarily poor performance. By arbitrary performance here we mean that the convergence time can be extended arbitrarily far into the future while still satisfying Assumption 2. We now discuss one such example.

**Example 1.** Consider the following 2-dimensional streaming regression problem

$$f_t(\boldsymbol{x}_t) = (1 - \boldsymbol{D}_t^\mathsf{T} \boldsymbol{x}_t)^2 \tag{39}$$

where $\boldsymbol{D}_t \in \mathbb{R}^2$ is the time varying data vector that only takes on two values (depending on the time $t$)

$$\boldsymbol{D}_t = \begin{cases} [1 \ \ 0]^\mathsf{T}, & \text{for } t < \tau \\ [0 \ \ 1]^\mathsf{T}, & \text{otherwise.} \end{cases} \tag{40}$$

The only fixed value that is optimal for all time in Example 1 is $\boldsymbol{x}^* = [1 \ \ 1]^\mathsf{T}$. Any algorithm (even an oracle with access to $\boldsymbol{D}_t$) that proposes updates at time $t$ given information at $t-1$ will have instantaneously poor performance at time $t = \tau$. Even though a fixed optimal solution exists, no algorithm can see this change coming. We can make this a little more tangible within the context of gradients by explicitly writing them down

$$\nabla f_t(\boldsymbol{x}_t) = \begin{cases} \begin{bmatrix} -2(1 - x_{1,t}) \\ 0 \end{bmatrix}, & \text{for } t < \tau \\[4mm] \begin{bmatrix} 0 \\ -2(1 - x_{2,t}) \end{bmatrix}, & \text{otherwise} \end{cases} \tag{41}$$

where we have used the following subscript notation

$$\boldsymbol{x}_t = \begin{bmatrix} x_{1,t} & x_{2,t} \end{bmatrix}^{\mathsf{T}}.$$

Using the gradients in Equation (41) to update $\boldsymbol{x}_t$ means that before $t = \tau$ only the first entry in $\boldsymbol{x}_t$ will be updated and the second entry in $\boldsymbol{x}_t$ will simply remain at its initial condition. Then at time $t = \tau$ when we calculate the cost $f_t$ there will be an instantaneous change that will no longer take into account all the learning that has occurred through the successive updates in $x_{1,t}$ as the new cost function from that point forward will only be a function of $x_{2,t}$. So if I want to arbitrarily increase the time to convergence in this setting one simply needs to increase $\tau$. This example is rather silly but demonstrates why convergence rates in the setting of time varying (convex) cost functions isn't really a thing that is studied without extra constraints (beyond convexity) being imposed on $f_t$.

## 7 Conclusions

The advantage of our approach is a natural setting for situations where learning is performed in real-time for arbitrary amounts of time. In our setting the learning rates do not need to asymptotically decrease over time. In fact for stability to hold the learning rates necessarily have to be bounded away from 0, see Equations (3b) and (3c). In addition, the structure of our Lyapunov candidates mirror those used in control, and could likely be extended to settings where there is feedback with a dynamical system (Gaudio et al., 2019). We recognize that no method or technique is a panacea. There are scenarios where the optimality conferred by Regret Analysis will be preferred, but we also think there are going to be growing applications in real-time scenarios where stability will be paramount. Ultimately, we simply hope that these methods might be useful in providing further insight into gradient descent with second order dynamics for time varying objective functions as well as **stochastic** gradient descent with second order dynamics as well. We are particularly interested in trying to extend these results to the stochastic setting and the accompanying relaxation of Assumption 2.

**Code Availability**

Code to reproduce the figures is available at [https://github.com/gibsonlab/stability_gd_2_tv](https://github.com/gibsonlab/stability_gd_2_tv)

**Author Contributions**

Following the CRediT taxonomy standard [https://credit.niso.org](https://credit.niso.org)

TEG: Conceptualization, Formal analysis, Funding acquisition, Investigation, Methodology, Project administration, Resources, Software, Supervision, Validation, Visualization, Writing – original draft, Writing – review & editing

SA: Formal analysis, Investigation, Methodology, Software, Validation, Visualization, Writing – review & editing

AP: Validation, Writing – review & editing

JEG: Conceptualization, Validation, Writing – review & editing

AMA: Conceptualization, Funding acquisition, Supervision, Writing – review & editing

**Acknowledgments**

This work was supported by the Boeing Strategic University Initiative (Annaswamy) and NIH R35GM143056 (Gibson)

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

# A  Technical Lemmas and Corollaries

These technical lemmas can be found in most standard texts on convex optimization (Boyd & Vandenberghe, 2004; Bubeck, 2015; Hazan, 2022)

**Lemma A.1** (First order condition of convexity (Boyd & Vandenberghe, 2004, §3.1.3)). *Let $f : \mathbb{R}^n \to \mathbb{R}$ be a differentiable convex function. Then $\forall \boldsymbol{x}, \boldsymbol{y} \in \mathbb{R}^N$,*

$$f(\boldsymbol{y}) \geq f(\boldsymbol{x}) + \nabla f(\boldsymbol{x})^\mathsf{T}(\boldsymbol{y} - \boldsymbol{x})$$

**Lemma A.2** (Basic property of smooth convex functions). *A continuously differentiable convex function $f : \mathbb{R}^N \to \mathbb{R}$ is $L$-smooth if and only if $\forall \boldsymbol{x}, \boldsymbol{y} \in \mathbb{R}^N$,*

$$f(\boldsymbol{y}) \leq f(\boldsymbol{x}) + \nabla f(\boldsymbol{x})^\mathsf{T}(\boldsymbol{y} - \boldsymbol{x}) + \frac{L}{2}\|\boldsymbol{y} - \boldsymbol{x}\|^2.$$

**Lemma A.3** (Basic property of strongly convex functions). *If a continuously differentiable convex function $f : \mathbb{R}^N \to \mathbb{R}$ is $\sigma$-strongly convex , then $\forall \boldsymbol{x}, \boldsymbol{y} \in \mathbb{R}^N$*

$$(\nabla f(\boldsymbol{x}) - \nabla f(\boldsymbol{y}))^\mathsf{T}(\boldsymbol{x} - \boldsymbol{y}) \geq \sigma\|\boldsymbol{x} - \boldsymbol{y}\|^2.$$

**Lemma A.4** (co-coercivity of smooth convex functions). *If a continuously differentiable convex function $f : \mathbb{R}^N \to \mathbb{R}$ is $L$-smooth , then $\forall \boldsymbol{x}, \boldsymbol{y} \in \mathbb{R}^N$*

$$(\nabla f(\boldsymbol{x}) - \nabla f(\boldsymbol{y}))^\mathsf{T}(\boldsymbol{x} - \boldsymbol{y}) \geq \frac{1}{L}\|\nabla f(\boldsymbol{x}) - \nabla f(\boldsymbol{y})\|^2.$$

**Lemma A.5** (co-coercivity of smooth and strongly convex functions). *If a continuously differentiable convex function $f : \mathbb{R}^N \to \mathbb{R}$ is $L$-smooth and $\sigma$-strongly convex , then $\forall \boldsymbol{x}, \boldsymbol{y} \in \mathbb{R}^N$*

$$(\nabla f(\boldsymbol{x}) - \nabla f(\boldsymbol{y}))^\mathsf{T}(\boldsymbol{x} - \boldsymbol{y}) \geq \frac{\sigma L}{\sigma + L}\|\boldsymbol{x} - \boldsymbol{y}\|^2 + \frac{1}{\sigma + L}\|\nabla f(\boldsymbol{x}) - \nabla f(\boldsymbol{y})\|^2.$$

**Lemma A.6** (distance to minimizer for smooth convex functions). *If a continuously differentiable convex function $f : \mathbb{R}^N \to \mathbb{R}$ is $L$-smooth , then $\forall \boldsymbol{x}, \boldsymbol{y} \in \mathbb{R}^N$*

$$\frac{1}{2L}\|\nabla f(\boldsymbol{x})\|^2 \leq f(\boldsymbol{x}) - f(\boldsymbol{x}^*) \leq \frac{L}{2}\|\boldsymbol{x} - \boldsymbol{x}^*\|^2.$$

**Lemma A.7** (distance to minimizer for strongly convex functions). *If a continuously differentiable function $f : \mathbb{R}^N \to \mathbb{R}$ is $\sigma$-strongly convex, then $\forall \boldsymbol{x}, \boldsymbol{y} \in \mathbb{R}^N$*

$$\frac{1}{2\sigma}\|\nabla f(\boldsymbol{x})\|^2 \geq f(\boldsymbol{x}) - f(\boldsymbol{x}^*) \geq \frac{\sigma}{2}\|\boldsymbol{x} - \boldsymbol{x}^*\|^2.$$

**Lemma A.8** ((Bubeck, 2015, Lemma 3.5)). *Let $f_t$ be a $L$-smooth convex function. Then $\forall \boldsymbol{x}, \boldsymbol{y} \in \mathbb{R}^N$,*

$$f(\boldsymbol{x}) - f(\boldsymbol{y}) \leq \nabla f_t(\boldsymbol{x})^\mathsf{T}(\boldsymbol{x} - \boldsymbol{y}) - \frac{1}{2L}\|\nabla f(\boldsymbol{x}) - \nabla f(\boldsymbol{y})\|^2.$$

**Corollary A.8.1.** *Under the setting in Lemma A.8 with $\boldsymbol{x} = \boldsymbol{x}_t$, $\boldsymbol{y} = \boldsymbol{x}^*$, and $\nabla f_t(\boldsymbol{x}^*) = 0$ the bound becomes*

$$\nabla f_t(\boldsymbol{x}_t)^\mathsf{T}(\boldsymbol{x}^* - \boldsymbol{x}_t) \leq [f_t(\boldsymbol{x}^*) - f_t(\boldsymbol{x}_t)] - \frac{1}{2L}\|\nabla f_t(\boldsymbol{x}_t)\|^2.$$

**Lemma A.9.** *Under the setting in Lemma A.8 with $\boldsymbol{x} = \boldsymbol{x}_t$, $\boldsymbol{y} = \boldsymbol{x}^*$, and $\nabla f_t(\boldsymbol{x}^*) = 0$ and for any $\lambda \in [0, 1]$ the bound becomes*

$$\nabla f_t(\boldsymbol{x}_t)^\mathsf{T}(\boldsymbol{x}^* - \boldsymbol{x}_t) \leq (1 - \lambda)[f_t(\boldsymbol{x}^*) - f_t(\boldsymbol{x}_t)] - \frac{1 + \lambda}{2L}\|\nabla f_t(\boldsymbol{x}_t)\|^2.$$

*Proof.* Add and subtract $\lambda[f_t(\boldsymbol{x}^*) - f_t(\boldsymbol{x}_t)]$ and apply Lemma A.6. ∎

# B Experimental Details

For the optimization function $f_t$ in (37) it is useful to note that the second derivative of

$$\frac{\partial^2}{\partial x} f_t(x) = b_t^2 \operatorname{sech}^2(b_t(-c_t + x)) \leq b_t^2$$

and thus $f_t$ is an $L_t$-smooth function where $L_t = b_t^2$. In all simulations the HT parameters are chosen as $\gamma_t = 1.5$, $\mu_t = 1$, $\beta_t = 1/1.5$ coinciding with Corollary 1.1.

## B.1 Details for Figure 1

The algorithms are optimizing Equation (37) with

- $a_t = 5$
- $b_t = 7$
- $c_t = 0$ for $t < 50$ and then $c_t = 5$ for $t \geq 50$

For GD, NAGD, and HT the learning rate normalizer is $N_t = N = 49$. All methods are initialized with $x_0 = y_0 = z_0 = 5$

## B.2 Details for Figure 2

The algorithms are optimizing Equation (37) with

- $a_t = 5$
- $b_t = 7$ for $t < 50$ and then $b_t = 21$ for $t \geq 50$
- $c_t = 0$

For GD, NAGD, the learning rate normalizer is $N = 49$ throughout. For TN-GD, TN-SAGD and HT $N_t = 49$ for $t < 50$ and then $N_t = 441$ for $t \geq 50$. All methods are initialized with $x_0 = y_0 = z_0 = 7$

## B.3 Details for Figure 3

The algorithms are optimizing Equation (37) with

- $a_t = 5$
- $b_t = 7$
- $c_t = 0$ for $t < 50$, $c_t = 5$ for $50 \leq t < 150$, $c_t = -4$ for $150 \leq t < 300$, and then $c_t = 0$ for $t \geq 300$

For HT the learning rate normalizer is $N_t = 49$. For Adam and Adagrad the learning rate $\alpha$ was manually tuned to have the best possible performance and was set to $\alpha = 1$.

