# OpenReview forum: "On the stability of gradient descent with second order dynamics for time-varying cost functions"
_TMLR — Accepted by TMLR_

### Review · Reviewer_ezmR · 2024-11-06

**Summary Of Contributions:**

In this work the authors analyzed gradient descent with second order dynamics when applied to explicitly time varying cost functions and provide more general stability guarantees. These results aid in the design and certification of optimization schemes to ensure safe and reliable deployment for real-time learning applications. The authors consider the general convex setting and strongly convex setting and proved the convergence results in these two different conditions. They also provide the empirical results from numerical experiments that are consistent with the theoretical analysis.

**Audience:**

Yes

**Broader Impact Concerns:**

There are no concerns on the ethical implications of the work that would require adding a Broader Impact Statement.

**Claims And Evidence:**

Yes

**Requested Changes:**

The authors must made changes or justify their work for all the four weaknesses mentioned in the Strengths And Weaknesses section.

The most important requested change is the weakness 4 about the proof of the non-asymptotic sublinear convergence rate of the high-order tuner methods for the general convex case. If the authors could prove the explicit convergence rate under the general convex case, I suggest that this submission should be accepted by TMLR. Otherwise, this submission does not deserve a publishment.

**Strengths And Weaknesses:**

This submission is organized with clear structure. The authors provided the detailed mathematical proof for all the theoretical analysis of the convergence results of the high-order tuner algorithms applied to the explicitly time varying cost functions. They also conduct numerical experiments and these empirical results validate the theoretical results. However, this submission has the following weaknesses:

1. The authors should present more background introduction for the high-order tuner methods that are the gradient descent methods with second order dynamics in section 2. This high-order tuner method is totally different from the standard gradient descent. Hence, it is reasonable for the authors to better introduce the previous related works and the illustrate more ideas behind this high-order tuner method.

2. This high-order tuner method has one issue that the parameter $\alpha_t$ depends on the parameter $N_t$, while $N_t$ must satisfies the condition that $N_t \geq L_t$ as specified in different theorems. However, $L_t$ is the smoothness parameter of the objective function. Hence, to achieve the different convergence results, we need to know these smoothness parameter $L_t$ of the objective function when we implement the high-order tuner method. This may cause some issues because in practice, we may not know or it is difficult to estimate the values of these $L_t$.

3. Proposition 1 and the warm up of Theorem 1. Hence, proposition 1 is a bit redundant in this submission. Maybe it can be moved to appendix. Also proposition 2 should be Theorem 2.

4. The most significant weakness of this submission is that the authors only prove asymptotic convergence results in Proposition 1 and Theorem 1 for the general convex case. While for the strongly convex case, the authors proved a non-asymptotic linear convergence rate as specified in proposition 2. For the vanilla GD, it can reach a linear convergence rate for strongly convex case while it can also reach a sub-linear convergence rate for the general convex case. Therefore, the authors need to investigate and prove a non-asymptotic explicit sub-linear convergence rate for the general convex condition. The current result of $\lim_{t \to \infty}f(x_t) = f(x_*)$ is not impressive enough.

---

> ### Author Response · Authors · 2024-12-20
>
> Our line by line response to your comments are below. Once a consensus is reached we will upload our revised manuscript.
>
> ---
> `Weakness:` 1. The authors should present more background introduction for the high-order tuner methods that are the gradient descent methods with second order dynamics in section 2. This high-order tuner method is totally different from the standard gradient descent. Hence, it is reasonable for the authors to better introduce the previous related works and the illustrate more ideas behind this high-order tuner method.
>
> `Answer:` In our revised manuscript we will add more history on the original work of Morse and their high-order tuner idea into the introduction.
>
> ---
> `Weakness:` 2. This high-order tuner method has one issue that the parameter $\alpha_t$ depends on the parameter $N_t$, while $N_t$ must satisfies the condition that $N_t \geq L_t$ as specified in different theorems. However, $L_t$ is the smoothness parameter of the objective function. Hence, to achieve the different convergence results, we need to know these smoothness parameter $L_t$ of the objective function when we implement the high-order tuner method. This may cause some issues because in practice, we may not know or it is difficult to estimate the values of these $L_t$.
>
> `Answer:`  This is indeed a weakness. In the revised manuscript we will mention this explicitly in the discussion and the fact that in practice one would need to run some kind of smoothness estimator for which their are many quasi-newton like methods to choose from.
>
> ---
> `Weakness:` 3. Proposition 1 and the warm up of Theorem 1. Hence, proposition 1 is a bit redundant in this submission. Maybe it can be moved to appendix. Also proposition 2 should be Theorem 2.
>
> `Answer:` We purposefully chose to make proposition 1 simple and to include it in the main text precisely because our proof technique and approach is not as mainstream as others in the ML community. So we wanted to hold the readers hand a little bit. We will keep Proposition 1 in the main text.
>
> It is a matter of taste whether to call Proposition 2 a theorem or a proposition and we will still call it a proposition because it uses the same assumptions and constraints as Proposition 1. It is also not the most general bound we can obtain for strongly convex functions (another reason why we called it a proposition). We discussed this at the beginning of section 3 in our original paper.
>
> >"In section §3.1 we present our first result for smooth convex functions (Proposition 1), with the more general results coming with Theorem 1 in §3.2 . We close this section with our analysis of strongly convex functions in §3.3 with Proposition 2. For the strongly convex setting obtaining the exponential convergence rate in a simple form necessitates some degree of simplification so we use the simplified setting of Proposition 1 in our presentation of Proposition 2." -- Beginning of section 3 in our paper.
>
> ---
> `Weakness:` 4. The most significant weakness of this submission is that the authors only prove asymptotic convergence results in Proposition 1 and Theorem 1 for the general convex case. While for the strongly convex case, the authors proved a non-asymptotic linear convergence rate as specified in proposition 2. For the vanilla GD, it can reach a linear convergence rate for strongly convex case while it can also reach a sub-linear convergence rate for the general convex case. Therefore, the authors need to investigate and prove a non-asymptotic explicit sub-linear convergence rate for the general convex condition. The current result of $\lim_{t \to \infty}f(x_t) = f(x_*)$ is not impressive enough.
>
> `Answer:` Our intent in providing the example in Section 6 was to preempt this line of criticism.  With Example 1 we show that no method can converge to $x^*$ faster than  $t=\tau$ and $\tau$ can be arbitrarily large. This means that in our setting with time-varying convex functions under the assumptions as we have provided there is no such thing as a non-asymptotic convergence rate.
>
> This same point of criticism was raised by reviewer @Wios so we  did a bad job getting our point across. Do you have any suggestions for how we can make this more clear? As it stands Example 1 is a proof (by counter example) that non-asymptotic convergence rates do not exist for time varying convex functions under our setting.

---

> > ### Comment · Reviewer_ezmR · 2025-01-16
> >
> > The authors addressed all my questions. I suggest to accept this submission.

---

> > > ### Comment · Action_Editor_wEVW · 2025-01-16
> > > **Official recommendation**
> > >
> > > Dear Reviewer ezmR,
> > >
> > > Thank you for your suggestion and comments. Would you please submit your official recommendation to TMLR? Thanks.
> > >
> > > Best regards,
> > > AE

---

### Review · Reviewer_1pi2 · 2024-12-02

**Summary Of Contributions:**

This paper develops a more general stability guarantee for gradient descent with second order dynamics when applied to time-varying cost functions. Specifically, they use better-designed hyper-parameters and a different Lyapunov candidate to build a less conservative bound on the hyperparameters, which do not require asymptotically decreasing learning rates over time.

**Audience:**

Yes

**Claims And Evidence:**

Yes

**Requested Changes:**

As mentioned above, the authors can add more explanations about the advantages of using $\\|\mathbf{z}_t-\mathbf{y}_t\\|$ instead of $\\|\mathbf{z}_t-\mathbf{x}_t\\|$.

**Strengths And Weaknesses:**

Strengths:
1. Compared with existing works, this paper allows a more flexible choice of hyperparameters in the gradient descent step, especially the learning rate. And they also use a different Lyapunov candidate.
2. For the strongly convex setting, they build the convergence of the Lyapunov function to zero instead of a compact set in other works.
3. Corollary 1.1 provides a practical choice of those hyper-parameters, which is useful in real applications.

Weaknesses:
1. After reading the paper, it is still a little unclear to me about the advantages of using $\\|\mathbf{z}_t-\mathbf{y}_t\\|$ instead of $\\|\mathbf{z}_t-\mathbf{x}_t\\|$, could the authors explain more about the benefits of using this different Lyapunov candidate?
2. As the authors stated, the HT methods cannot perform as well as the Regret Analysis in some scenarios. Considering the conditions under which the HT methods can perform better would be interesting.
3. Assumption 2 may not always hold for real applications. In other words, it would be more appropriate to consider time-varying optimal $x$ in the setting of time-varying cost functions.

---

> ### Author Response · Authors · 2024-12-20
>
> Our line by line response to your comments are below. Once a consensus is reached we will upload our revised manuscript.
>
> ---
> `Weakness:` 1. After reading the paper, it is still a little unclear to me about the advantages of using $|| \mathbf{z}_t-\mathbf{y}_t||$ instead of $|| \mathbf{z}_t-\mathbf{x}_t||$, could the authors explain more about the benefits of using this different Lyapunov candidate?
>
> `Answer:` Sure. When you have $||z_t-y_t||$ then in your next Lyapunov step you will have  $|| z_{t+1}- y_{t+1}||$ whose components are
> $$\\begin{align}
> z_{t+1} = z_t-\eta_t\nabla f_t(x_t)  \\\\  y_{t+1} = x_t-\alpha_t\nabla f_t(x_t)
> \\end{align}$$
> which can be easily reduced in the downstream steps. If you begin with $||z_t-x_t||$ then you must contend with $||z_{t+1}-x_{t+1}||$ whose components are
> $$
> z_{t+1} = z_t-\eta_t\nabla f_t(x_t)$$
> $$ \\begin{align}
> x_{t+1} &= \beta_{t+1} y_{t+1} + (1-\beta_{t+1}) z_{t+1} \\\\
>          &= \beta_{t+1} (x_t-\alpha_t\nabla f_t(x_t))+ (1-\beta_{t+1})(z_t-\eta_t\nabla f_t(x_t) \\end{align}
> $$
> Having the extra $\beta_{t+1}$ terms floating around makes things difficult because later you will have $\beta_t$ terms as well. Then you end up with products or ratios involving $\beta_t$ an $\beta_{t+1}$ and you have to worry about the sequence of $\beta_t$ itself. In prior work they just assumed $\beta_t$ was a constant so they never had to deal with this issue, but in our more generall setting it was significantly easier to contend with $||\mathbf{z}_t-\mathbf{y}_t||$.
>
> ---
> `Weakness:` 2. As the authors stated, the HT methods cannot perform as well as the Regret Analysis in some scenarios. Considering the conditions under which the HT methods can perform better would be interesting.
>
> `Answer:` We did consider the conditions under which HT performs better. We think you meant say "the conditions under which regret based schemes perform better"? We agree that it would be interesting to explore where regret based schemes perform better but that would be out of scope for this paper
>
> In our closing we stated
> >"We recognize that no method or technique is a panacea. There are scenarios where the optimality conferred by Regret Analysis will be preferred, but we also think there are going to be growing applications in real-time scenarios where stability will be paramount."
>
> to make it clear that we were not implying that our analysis is the best and only way to analyze gradient descent
>
> ---
> `Weakness:` 3. Assumption 2 may not always hold for real applications. In other words, it would be more appropriate to consider time-varying optimal  in the setting of time-varying cost functions.
>
> `Answer:` As discussed in our response to reviewer @Wios. We will add a theorem or an extended discussion regarding stability when there are a finite number of changes to $x^*$ to coincide with our simulation experiments. We gave the sketch of such a proof in our response to @Wios.

---

> > ### Comment · Reviewer_1pi2 · 2025-01-02
> >
> > Thanks for the further clarification. The authors have addressed my concerns, so I recommend accepting the paper.

---

### Review · Reviewer_Wios · 2024-12-17

**Summary Of Contributions:**

This paper analyzes gradient descent methods with second-order dynamics for time-varying objective functions through the lens of stability analysis. The paper considers a Lyapunov/potential function framework that shows stability (in the sense of a bounded trajectory) with time-varying objectives for an algorithm that resembles Nesterov acceleration (though the analysis technically does not recover Nesterov's accelerated gradient method). The authors provide analysis in both the smooth and strongly convex settings with relaxed hyperparameter constraints compared to prior work. Finally, the paper provides some experimental verification for the stability of the iterates under time-varying objectives.

**Audience:**

Yes

**Broader Impact Concerns:**

N/A.

**Claims And Evidence:**

Yes

**Requested Changes:**

- Please address my points in the strengths and weaknesses sections.
- Generally: what is particularly difficult about the time-varying objective when you assume overparameterization? The analysis looks somewhat standard and similar to existing Lyapunov analysis (or "potential function" in general) of optimizationa algorithms (e.g. [1, 2]). Can you elaborate on this here?

[1] Bansal, Nikhil, and Anupam Gupta. "Potential-function proofs for gradient methods." Theory of Computing 15.1 (2019): 1-32.
[2] Wilson, Ashia C., Ben Recht, and Michael I. Jordan. "A Lyapunov analysis of accelerated methods in optimization." Journal of Machine Learning Research 22.113 (2021): 1-34.

**Strengths And Weaknesses:**

- (Strength) The paper is written clearly, the analysis is more general than prior work (at least when confined to the setting of time-varying objectives).
- (Strength) The paper provides interesting connections between optimization and control theory proofs.
- (Weakness) Several technical issues in the proofs need to be addressed:
1. The paper states that "for ∆Vt+1 ≤ 0 we need c1, c4 and $c_2^2 - 4c_1c_3$ to be less than or equal to zero", but this is incorrect - the condition should be $4c_1c_3 - c_2^2 \leq 0$. While this doesn't seem to affect the final result since the proof sets this quantity to exactly 0, the statement should be fixed.
2. The claim that "the optimal point can actually change over time and stability is maintained" appears too strong. The paper assumes $x^*$ is fixed for proving asymptotic convergence but claims stability holds even with time-varying $x_t^*$. I don't see how the proof can be modified to handle this, it relies crucially on $f_t (x_t) - f(x^*) \leq 0$ for all $t$.
3. The claim that "In the stochastic setting Assumption 2 is equivalent to the unbiased sub-gradient assumption"-- I honestly can't see why this is the case. Could you clarify?
- (Weakness) The paper essentially assumes overparameterization ($x^*$ minimizes all the $f_t$) but doesn't exploit this to potentially get better rates. Specifically, the paper claims it's impossible to have learning rates scaling beyond $\mathcal{O}(1/N_t)$ for time-varying objectives, but under overparameterization this limitation may not hold. At least the authors should provide a lower bound showing this. Limiting learning rates to be this small is why the analysis here can't recover Nesterov's method.

---

> ### Author Response · Authors · 2024-12-19
>
> Our line by line response to your comments are below. Once a consensus is reached we will upload our revised manuscript.
>
> ---
> `Weakness:` 1. The paper states that "for ∆Vt+1 ≤ 0 we need c1, c4 and $c_2^2 - 4c_1c_3$ to be less than or equal to zero", but this is incorrect - the condition should be $4c_1c_3 - c_2^2 \leq 0$. While this doesn't seem to affect the final result since the proof sets this quantity to exactly 0, the statement should be fixed
>
> `Answer:` We believe the condition as stated in the paper is correct. After completing the square in (10)  the following inequality needs to hold
> $$
> c_3- \frac{c_2^2}{4c_1} \leq 0
> $$
> If we multiply by $c_1$ **(which is a negative number meaning the direction of the inequality switches)** we obtain the following
> $$
> 4c_1c_3-c_2^2 \geq 0
> $$
> If we multiply by -1 flipping the inequality again and rearranging terms on the left hand side we have
> $$
> c_2^2-4c_1c_3 \leq 0
> $$
> as stated in the paper.
>
> ---
> `Weakness:` 2. The claim that "the optimal point can actually change over time and stability is maintained" appears too strong. The paper assumes $x^*$ is fixed for proving asymptotic convergence but claims stability holds even with time-varying $x_t^*$. I don't see how the proof can be modified to handle this, it relies crucially on $f_t (x_t) - f(x^*) \leq 0$ for all $t$.
>
> `Answer:`  **We should have been more explicit about why this holds and included another theorem or at the very least an extended discussion about why this holds.** In our proof setting we make no assumptions about the initial conditions of the parameters or any other a priori assumptions about their boundedness and that is why we concluded that we can handle time varying $x_t^*$. Here is an informal proof.
>
> Consider our problem setting with arbitrary initial conditions $x_0,y_0,z_0$  and an optimal $x^*_{[1]}$ to begin with. Our proof says that we are stable and we will asymptotically converge to $x^*_{[1]}$. Now at some time  $T$ we get a new optimal  $x^*_{[2]}$ (with the subscript $[2]$). Importantly we do nothing to the algorithm and it just continues to run as originally defined. For analysis we now say $x_T,y_T,z_T$ are our new initial conditions. Recall that our proof explicitly states that we are stable for arbitrary initial conditions and so we will still be bounded for all time with the new $x^*_{[2]}$ and will now asymptotically converge to the new $x^*_{[2]}$ as well. This informal proof can be made more formal by then restricting the number of times $x^*$ switches to be finite. One can go further and prove uniform stability with an infinite number of switches but this would require lots of extra machinery and would explicitly depend on the sequence of $f_t$ and $x^*_t$, so we wont be addressing that more complicated scenario in this work.
>
> This notion of stability never enters one's mind when studying gradient algorithms with regret analysis because you a priori assume all the parameters and gradients are uniformly bounded before you even start the analysis. In our revised paper we will have a meta-theorem either in the main text or in the supplement along with an extended discussion about what is typically assumed when studying gradient descent within the context of regret analysis vs what is assumed within our control theoretic approach and why proving boundedness with arbitrary initial conditions is useful.
>
> ---
> `Weakness:` 3. The claim that "In the stochastic setting Assumption 2 is equivalent to the unbiased sub-gradient assumption"-- I honestly can't see why this is the case. Could you clarify?
>
> `Answer:` We will remove this claim from the paper and leave any connections to the stochastic setting to a future paper.
>
> ---
>
> `Weakness:` The paper essentially assumes overparameterization ($x^*$ minimizes all the $f_t$) but doesn't exploit this to potentially get better rates. Specifically, the paper claims it's impossible to have learning rates scaling beyond  $\mathcal O(1/N_t)$ for time-varying objectives, but under overparameterization this limitation may not hold. At least the authors should provide a lower bound showing this. Limiting learning rates to be this small is why the analysis here can't recover Nesterov's method.
>
> `Answer:` Our intent in providing the example in Section 6 was to preempt this line of criticism.  With Example 1 we show that no method can converge to $x^*$ faster than  $t=\tau$ and $\tau$ can be arbitrarily large. This means that in our setting with time-varying convex functions under the assumptions as we have provided there is no such thing as a non-asymptotic convergence rate.  This same point of criticism was raised by another reviewer so we  did a bad job getting our point across. Do you have any suggestions for how we can make this more clear? As it stands Example 1 is a proof (by counter example) that non-asymptotic convergence rates do not exist for time varying convex functions under our setting.

---

> ### Author Response · Authors · 2024-12-19
>
> `Weakness:` - Generally: what is particularly difficult about the time-varying objective when you assume overparameterization? The analysis looks somewhat standard and similar to existing Lyapunov analysis (or "potential function" in general) of optimizationa algorithms (e.g. [1, 2]). Can you elaborate on this here?
>
> [1] Bansal, Nikhil, and Anupam Gupta. "Potential-function proofs for gradient methods." Theory of Computing 15.1 (2019): 1-32. [2] Wilson, Ashia C., Ben Recht, and Michael I. Jordan. "A Lyapunov analysis of accelerated methods in optimization." Journal of Machine Learning Research 22.113 (2021): 1-34.
>
> `Answer:` In [1] their potential functions are either simply $||x_t-x^*||$ or they incorporate $f$ (which for us would be $f_t$).  The function $||x_t-x^*||$  alone does not work in our setting and in Remark 2 we discussed why $f_t$ can not be part of a Lyapunov function.
>
> >"Another subtle point here is that $f_t(x_{t}) - f_t(x^*)$ actually can not be included in the Lyapunov function as is common when analyzing time invariant cost functions ... A Lyapunov function has to be lowerbounded by a time-invariant non-decrescent function of the parameter you want to show is bounded. For instance if you have a time varying Lyapunov candidate $V_t(x)$ you will need to show that there exists a non-decrescent function $\alpha$ where $\alpha(0)=0$ and $0<\alpha(||{x}||) \leq V_t(x)$  for all $x \neq   0$ [Theorem 1, Kalman 1960]. With the cost function $f_t$ now explicitly varying over time the above condition can easily be violated without adding additional constraints on $f_t$." - Remark 2 in our paper
>
>
> With regards to [2] we see the same style of Lyapunov functions combining $||x_t-x^*||$ and $f(x_t)-f(x^*)$ [Equation (10) in [2]]. At the end of Section 4.3 we discuss the fact that when analyzing nesterov's method you will typically use $f(x_t)-f(x^*)$ in the Lyapunov candidate and made a call back to Remark 2 and why we cant follow this approach. In our original paper we only cited Ahn and Sra (2022) when doing this. We will now cite [1,2] in that same section as well. The challenge arises precisely because one can no longer use $f$ in the Lyapunov function.

---

> > ### Comment · Reviewer_Wios · 2025-01-02
> >
> > 1. Thanks for the clarification, I think it'd be potentially useful to point out, explicitly, which constants $c_i$ are positive and which are not.
> >
> > 2. I agree with your argument, but as you point out, it'd have to depend on either the number of switches or some other notion of switching cost (e.g. the squared norm of the difference, like the literature on dynamic regret).
> >
> > 3. The problem with the argument of Example 1 is that it just says we cannot have non-asymptotic convergence rates in general; But says nothing about the specific rates we can achieve when we can have non-asymptotic convergence rates. For example, the loss spike in here is bounded by the switching cost. I agree, however, that this is an extra assumption and that my original criticism does not apply. As for why we missed it.. I think it might just be because the section is placed at the very end of the paper. It's easy to miss there.
> >
> > 4. Thanks for the clarification.
> >
> > I recommend the paper for acceptance. The authors have adequately addressed all of my concerns.

---

### Author Response · Authors · 2024-12-19

Thank you for your thorough and detailed reviews. We will now begin with our rebuttal to the initial reviews. After a brief back and forth we will upload our revised manuscript.

---

### Author Response · Authors · 2025-01-10

A revised manuscript has now been uploaded. Major text changes in red. Most significant change is the new section 3.4 on time varying optimal points.

---

### Decision · Action_Editor_wEVW · 2025-01-18

**Recommendation:** Accept as is

**Comment:**

The paper substantially generalizes the existing stability analysis of gradient methods by allowing a more flexible choice of hyperparameters in the gradient descent step and time-varying functions. The analysis shows an interesting connection between optimization and control theory proofs. All the reviewers acknowledge the interests of the results and the strength of the theoretical analysis. The discussions and revision address the issues well.

**Audience:**

The stability and robustness of gradient descent methods are important for the deployment of models in real-time and safety critical systems. The analysis shows the connection between control theory and optimization. The derived results are interesting to the TMLR community.

**Claims And Evidence:**

The paper studies the stability of gradient descent methods with second-order dynamics to optimize time-varying cost functions, where the stability means that the algorithm has a bounded trajectory. The paper presents solid mathematical analysis with clear proofs. Experimental results are also presented to verify the stability of gradient descent with second-order dynamics.